# Longitudinal awake imaging of mouse deep brain microvasculature with super-resolution ultrasound localization microscopy

Yike Wang[1,2,3], Matthew R Lowerison[1,2,3], Zhe Huang[1,3], Qi You[1,4], Bing-Ze Lin[1,2,3], Daniel A Llano[1,5,6], Pengfei Song[1,2,3,4]*

[1]Beckman Institute for Advanced Science and Technology, University of Illinois Urbana-Champaign, Urbana, United States; [2]Department of Electrical and Computer Engineering, University of Illinois Urbana-Champaign, Urbana, United States; [3]Department of Biomedical Engineering, Duke University, Durham, United States; [4]Department of Bioengineering, University of Illinois Urbana-Champaign, Urbana, United States; [5]Department of Molecular and Integrative Physiology, University of Illinois Urbana-Champaign, Urbana, United States; [6]Neuroscience Program, University of Illinois Urbana-Champaign, Urbana, United States

*For correspondence:
pengfei.song@duke.edu

Competing interest: The authors declare that no competing interests exist.

## eLife Assessment

This study presents **important** methodologies for repeated brain ultrasound localization microscopy (ULM) in awake mice and a set of results indicating that wakefulness reduces vascularity and blood flow velocity. The data supporting these findings are **solid**. This study is relevant for scientists investigating vascular physiology in the brain.

**Abstract** Ultrasound localization microscopy (ULM) is an emerging imaging modality that resolves microvasculature in deep tissues with high spatial resolution. However, existing preclinical ULM applications are largely constrained to anesthetized animals, introducing confounding vascular effects such as vasodilation and altered hemodynamics. As such, ULM quantifications (e.g., vessel diameter, density, and flow velocity) may be confounded by the use of anesthesia, undermining the usefulness of ULM in practice. Here, we introduce a method to address this limitation and achieve ULM imaging in awake mouse brain. Pupillary monitoring was used to support the presence of the awake state during ULM imaging. Vasodilation induced by isoflurane was observed by ULM. Upon recovery to the awake state, reductions in vessel density and flow velocity were observed across different brain regions. In the cortex, the effects induced by isoflurane are more pronounced on venous flow than on arterial flow. In addition, serial in vivo imaging of the same animal brain at weekly intervals demonstrated the highly robust longitudinal imaging capability of the proposed technique. The consistency was further verified through quantitative analysis on individual vessels, cortical regions of arteries and veins, and subcortical regions. This study demonstrates longitudinal ULM imaging in the awake mouse brain, which is crucial for many ULM brain applications that require awake and behaving animals.

## Introduction

Sensitive imaging of correlates of activity in the awake brain is fundamental for advancing our understanding of neural function and neurological diseases. In the field of neuroscience, there is a growing interest in utilizing imaging techniques to study the rodent brain, which serves as a valuable model for investigating brain function (*Misgeld and Kerschensteiner, 2006*; *Ji et al., 2016*). Brain imaging modalities for rodents encompass a wide range of techniques, including but not limited to functional magnetic resonance imaging (fMRI) (*Grandjean et al., 2020*; *Gutierrez-Barragan et al., 2022*), positron emission tomography (PET) (*Toyama et al., 2005*; *Miranda et al., 2019*), one- and two-photon imaging (*Ma et al., 2016*; *Zong et al., 2021*), photoacoustic imaging (*Yao et al., 2015*; *Pang et al., 2021*), and more recently ultrasound localization microscopy (ULM) (*Errico et al., 2015*). ULM is uniquely capable of imaging microvasculature situated in deep tissue (e.g., at a depth of several centimeters). ULM can also be combined with the principles of functional ultrasound (fUS) (*Macé et al., 2011*; *Mace et al., 2013*; *Deffieux et al., 2021*; *Edelman and Macé, 2021*) to image whole-brain neural activity at a microscopic scale (*Renaudin et al., 2022*). The structural and functional imaging capabilities of ULM have opened new doors for numerous basic research and clinical applications that involve cerebral microvasculature (*Lowerison et al., 2022*; *Chavignon et al., 2022*; *Demené et al., 2021*).

At present, a key limitation associated with existing ULM brain imaging studies is the use of anesthesia, which induces profound alterations to cerebral blood flow (CBF), including changes in vessel size (e.g., diameter) and flow velocity (*Sullender et al., 2022*). As such, CBF measurements under anesthesia do not reflect the blood flow under the normal physiological state of the brain. In addition, anesthetics also have a significant attenuating effect on neural responses to sensory inputs, thereby impacting the neurovascular coupling process (*Masamoto and Kanno, 2012*; *Slupe and Kirsch, 2018*; *Reimann and Niendorf, 2020*; *Aksenov et al., 2015*; *Pisauro et al., 2013*; *Desai et al., 2011*). Therefore, in neuroscience research, brain imaging in the awake state is often preferred over imaging under anesthesia (e.g., fluorescence imaging *Flusberg et al., 2008*, photoacoustic imaging *Guo et al., 2021*; *Tang et al., 2015*, fMRI *Gutierrez-Barragan et al., 2022*, PET *Miranda et al., 2019*, and fUS *Bertolo et al., 2021*; *Macé et al., 2018*; *Sans-Dublanc et al., 2021*; *Brunner et al., 2021*). As ULM is gaining traction in many preclinical brain imaging applications, enabling ULM for awake animals has become crucial to eliminate the confounding vascular effects of anesthetics and obtain accurate structural and functional cerebrovascular measurements.

Another challenge associated with preclinical ULM brain imaging is to conduct long-term, longitudinal studies, which are essential for tracking disease progression or therapeutic impacts for many neurological disease applications (*Beckmann et al., 2019*; *Wang et al., 2021*; *Tournissac et al., 2022*). The key technical challenge for longitudinal ULM brain imaging is to find the same imaging plane and reconstruct consistent ULM images across different imaging sessions. Misalignment of imaging planes or tissue movement will undermine ULM imaging quality and result in inconsistent cerebrovascular quantifications. Although intact skull imaging has been shown to be feasible for short-term studies (e.g., over a few days) (*Chavignon et al., 2022*), long-term monitoring can be challenging because changes in skull properties over time (e.g., thickness and composition) could negatively impact ULM imaging quality. Currently, there is a strong need for methodological developments to enable longitudinal brain imaging with ULM.

In this study, we developed a method for awake and longitudinal ULM brain imaging in a mouse model to eliminate the confounding vascular effects of anesthesia and enable monitoring of cerebral vasculature over a 3-week period in the same animal. We constructed a head-fixed awake imaging platform and established a ULM image reconstruction metric to allow comparisons of ULM images acquired at different states of wakefulness (e.g., awake vs. anesthesia). Our method allowed detailed comparisons of local and global variations of the cerebral vasculature and blood flow under awake and anesthesia conditions. A detailed quantitative analysis of vessel diameter and blood flow velocity was performed. We also demonstrated robust longitudinal ULM brain imaging on the same animals with high repeatability over multiple weeks.

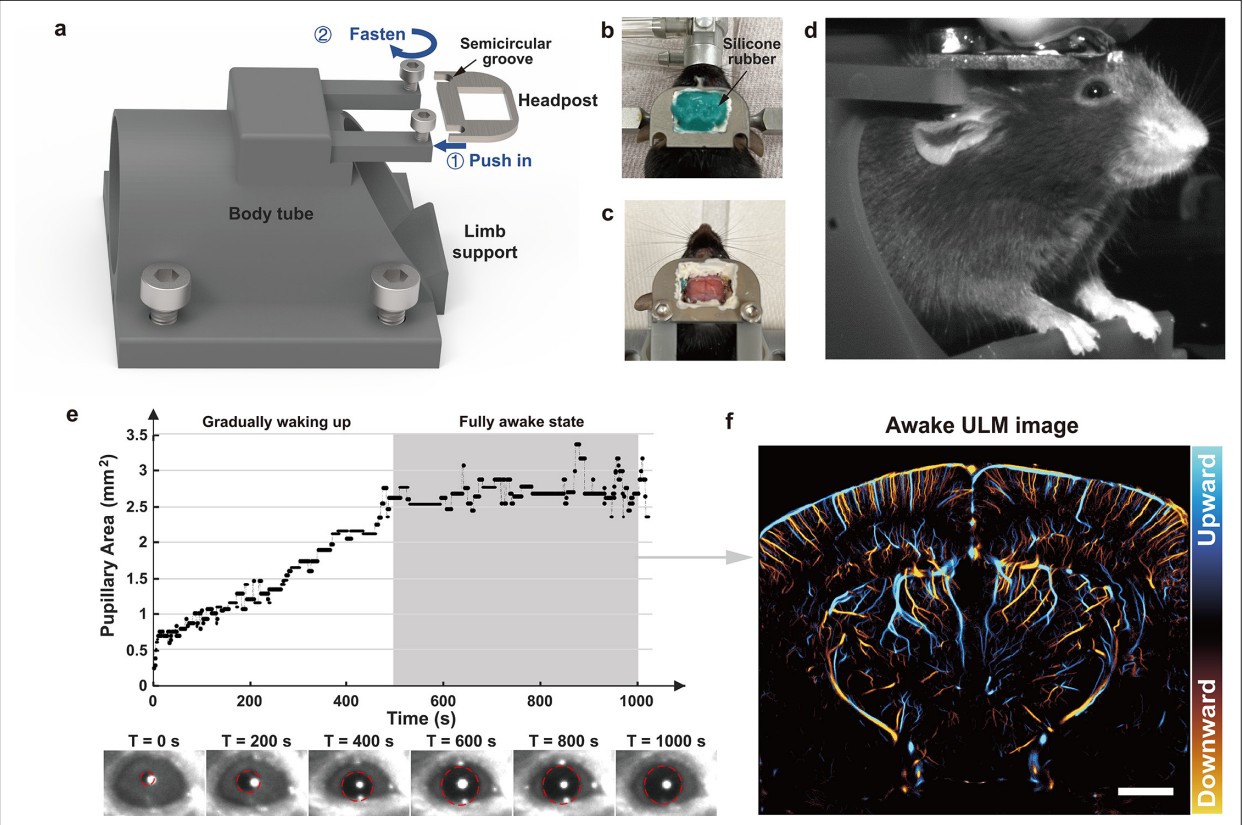

**Figure 1.** Experimental setup for awake mice ultrasound localization microscopy (ULM) brain imaging. (**a**) 3D model of the body tube enabling rapid fixation of the headpost onto the tube. (**b**) Top view photograph of the mouse after cranial window surgery, with the headpost protected by silicone rubber. (**c**) Photograph taken after the post-surgery recovery, with the removal of the silicone rubber protection. (**d**) Image captured by the camera positioned in front of the mouse during the imaging session. (**e**) Change of the pupillary area over time, after microbubble injection ($T = 0$ s), along with magnified portion of the pupil photo in d at six time points ($T = 0, 200, 400, 600, 800$, and $1000$ s). Pupil is outlined with a red dashed circle. (**f**) Awake ULM image obtained from data collected during a fully awake state, as indicated by the gray shaded area in (**e**) (scale bar: 1 mm).

The online version of this article includes the following figure supplement(s) for figure 1:

**Figure supplement 1.** Examples of poor imaging outcomes.

## Results

### ULM brain imaging can be performed in head-fixed awake mice

To achieve consistent ULM brain imaging while allowing limited movement in awake animals, a head-fixed imaging platform with a chronic cranial window was used in this study. Pre-surgery handling and post-surgery habituation were employed to alleviate animal stress and facilitate awake imaging (*Brunner et al., 2021*). To further minimize tissue motion, a 3D-printed body tube was adopted (*Brunner et al., 2021*) for animal immobilization (*Figure 1a*). A headpost was implanted during cranial window surgery to fix the animal's head to the body tube. In this procedure, the skull was removed, and the brain tissue was protected using a polymethyl pentene (PMP) membrane (*Sieu et al., 2015*), which was further protected by a layer of silicone rubber (*Figure 1b*). Animals were imaged after 1 week of surgical recovery and head-fixation habituation. Prior to each imaging session, the silicone rubber was gently removed with forceps, and the transparent PMP membrane allowed detailed examination of the brain surface to ensure absence of tissue damage (*Figure 1c*).

Contrast-enhancing microbubbles (MBs) were administered intravenously via tail vein in this study (details provided in Methods). Isoflurane anesthesia was terminated right after MB injection, marking the beginning of the data acquisition (i.e., $T = 0$ s). Mice were allowed to gradually regain consciousness. Throughout the awakening process, an infrared (IR) camera was used to monitor the pupil to provide a reference for the state of arousal (*Turner et al., 2023*; *Figure 1d*). *Figure 1e* shows the change in pupillary area over time, which reveals a gradual enlargement of the pupil after the cessation

of anesthesia. Due to differences in tail vein injection timing and anesthesia depth, the time required for each mouse to fully awaken varied. Although it was not feasible to get pupil size stabilized just after 500 s for each animal, ULM reconstruction only used the data that was acquired after the animal reached full pupillary dilation, to ensure that ULM accurately captures the cerebrovascular characteristics in the awake state. Ultrafast plane wave imaging data were collected for ULM reconstruction (details provided in Methods) and a representative awake ULM brain image is presented in *Figure 1f*.

## MB count serves as a quantitative metric for awake ULM image reconstruction

Due to the stochastic nature of MB localization and fluctuations of MB concentration in the blood stream, it is challenging to compare ULM images acquired at different states of wakefulness with different CBF conditions. Therefore, to obtain complete, fully saturated ULM images under different physiological conditions and MB concentrations, we used MB count as a measure of vessel saturation to determine the completeness of ULM reconstructions. To ensure high-quality ULM imaging, only MBs that were successfully tracked by the uTrack algorithm (details provided in Methods) were considered as effective MB signals that were utilized for ULM image quantification.

To facilitate equitable comparison of brain perfusion at different states, a practical saturation point enabling stable quantification of most vessels needs to be established. Our observations indicated that when the cumulative MB count reached 5 million, ULM images achieved a relatively stable state. Accordingly, in this study, the saturation point was defined as a cumulative MB count of 5 million. There are also possible alternatives for ULM image normalization. For example, different ULM images can be normalized to have the same saturation rate. However, the proposed method of using the same number of cumulative MB count for normalization enables the analysis of blood flow distribution across different brain regions from a probabilistic perspective. The following analysis uses this criterion.

*Figure 2a* compares ULM directional vessel density maps and flow speed maps generated with 1, 3, 5, and 6 million MBs, using the same animal as shown in *Figure 1*. To quantitatively confirm saturation, multiple vessel segments were selected for further analysis. *Figure 2b* presents the measured vessel diameter for a specific segment at various MB counts. After binarizing the ULM map, the vessel diameter was measured by calculating the distance from the vessel centerline to the edge. Each point along the centerline of the segment provided a diameter measurement, enabling calculation of the mean and standard error. At low MB counts, vessels appeared incompletely filled, leading to inaccurate estimation of vessel diameter due to incomplete profiles. For example, at 1–2 million MBs, the binarized ULM map displayed a width of only one or two pixels along the segment. As a result, the measurements always yielded the same diameter values (two pixels, ~10 um) with a consistently low standard error of the mean across the entire segment. With increased MB counts, the measured vessel diameter gradually rose, ultimately reaching saturation. The plots in *Figure 2b* show that vessel diameter stabilized at 5 million MB count. Additionally, *Figure 2c* illustrates the changes in flow velocity measured at different cumulative MB counts. The violin plots display the distribution of flow speed estimates for all valid centerline pixels within the selected segment. At low MB counts (1–3 million), flow velocity estimates fluctuated, but they stabilized as the MB count increased (4–6 million MBs). At 5 million MBs, flow velocity estimates were nearly identical to those at 6 million MBs, corroborating previous findings that vessel velocity measurements stabilize as MB count grows (*Hingot et al., 2019*). To assess the generalizability of the 5 million MB saturation condition, vessel segments from three different mice across various brain regions were examined. The results, shown in *Figure 2—figure supplement 1*, confirm that this saturation criterion applies broadly. Although the 5 million MB threshold may not ensure absolute saturation for all vessels, it is generally effective for vessels larger than 15 μm. This MB count threshold was therefore adopted as a practical criterion.

To further verify that the proposed MB bolus injection method can help to achieve ULM image saturation shortly after mice awaken from anesthesia, an analysis on the change in MB concentration over time was conducted once pupil size had stabilized ($T = 500$ s). *Figure 2d* shows a clear trend of decreasing MB concentration in the blood stream and increasing cumulative MB count with time. *Figure 2e* demonstrates a flattened vessel saturation curve and a rapidly reducing vessel filling rate, which is typical for ULM reconstruction (*Hingot et al., 2019*). The pixel filling rate dropped below 5% of the initial rate after 300 s of ULM data accumulation ($T = 800$ s), indicating ULM image saturation.

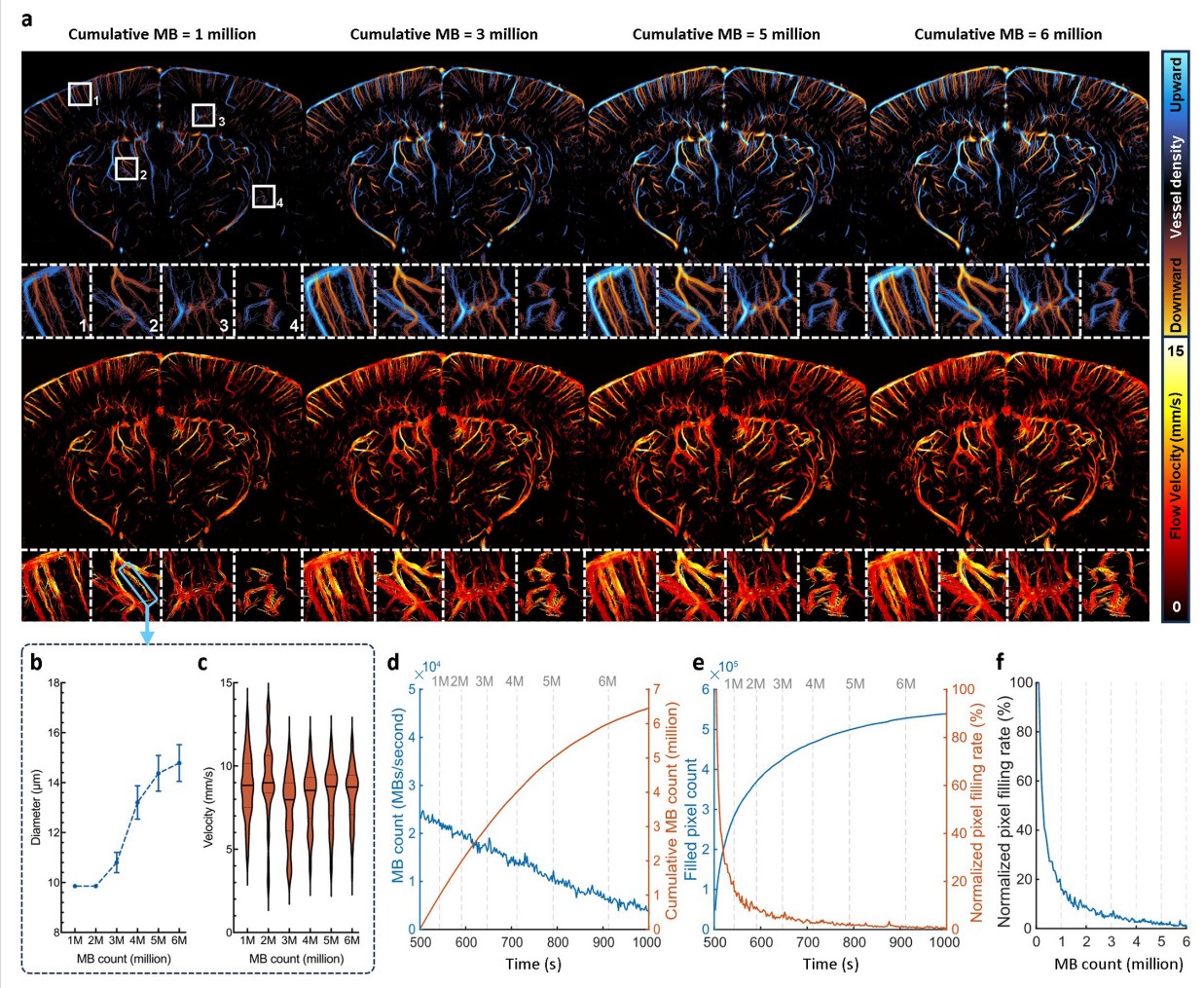

**Figure 2.** Data processing standards for awake mice ultrasound localization microscopy (ULM) imaging. (**a**) ULM directional vessel density maps and flow velocity maps at cumulative microbubble (MB) count of 1, 3, 5, and 6 million. (**b**) Vessel diameter measurements along the selected segment at varying cumulative MB counts. Each point along the vessel centerline provides a measurement, with the mean and standard error of the mean (SEM) plotted to show average diameter and variability. (**c**) Flow velocity measurements along the selected segment at varying cumulative MB counts. Each non-zero centerline pixel provides a velocity value, with the distribution across positions shown as a violin plot. (**d**) Time courses of MB count in each second (blue curve) and the cumulative MB count (orange curve) starting from $T = 500$ s. The vertical gray dashed lines indicate the time points when the cumulated MB count reaches 1, 2, 3, 4, 5, and 6 million. (**e**) Time courses of saturation level of the ULM image (blue curve) and the filling rate of pixels (orange curve). The filling rate is calculated by taking the derivative of the filled pixel count, and then normalized to the initial filling rate at the beginning of ULM reconstruction ($T = 500$ s). (**f**) Relationship between pixel filling rate and cumulative MB count, eliminating the time axis by plotting the orange curves from d and e together. This figure presents a case study based on the same mouse shown in *Figure 1*. The *x*-axis for (**d–f**) begins at 500 s because, at this point, the mouse's pupil size stabilized, indicating it had recovered to an awake state. Consequently, ULM images were accumulated starting from this time. It is important to note that not every mouse requires 500 s to fully awaken; the time to reach a stable awake state varies across individual mice.

The online version of this article includes the following figure supplement(s) for figure 2:

**Figure supplement 1.** Perfusion under the 5 million saturation standard for various small vessels.

---

*Figure 2f* further examines the relationship between the pixel filling rate and the cumulative MB count, which is independent of data acquisition time. The pixel filling rate at 5 million cumulative MBs is always below 5% of the initial rate for each experiment, ensuring image saturation. In summary, all the quantitative measurements indicate that ULM images obtained using the proposed metric (i.e., 5 million MBs) were complete and can be used to consistently measure cerebral vascular properties such as vessel diameter and blood flow velocity. All subsequent ULM images in this study were produced using this criterion.

## ULM reveals an increase in blood flow induced by isoflurane anesthesia

*Figure 3* presents a comparison of ULM directional vessel density maps and flow velocity maps in awake and anesthetized states for three different coronal planes from three animals. Four regions of interest (ROIs) were selected within each imaging plane to provide detailed comparisons. When comparing vessel density maps, ULM images that were acquired in the awake state demonstrate a global reduction of vessel density. The reduction is also clearly observed in magnified local regions, especially for region 5, which encompasses the pretectal region. ULM also reveals decreased vessel diameter which reflects vasoconstriction after the animals woke up (e.g., white arrows in regions 1, 3, and 6, corresponding to the thalamus, cortex, and midbrain/cortex overlap region). Focusing on ULM flow speed maps, a global reduction in flow speed can be clearly observed when transitioning from anesthetized to awake. Regional maps further revealed the significant flow speed reduction for most of the vessels throughout the brain (e.g., blue arrows).

To quantitatively compare the results from *Figure 3*, we performed various analyses in *Figure 4*. Since previous studies have shown that isoflurane induces different effects on arteries and veins (*Sullender et al., 2022*; *Rakymzhan et al., 2021*; *Cao et al., 2017*), we first selected a cortical artery and vein for comparison. In *Figure 4a*, a cortical region near region 3 from *Figure 3* was selected, and one artery and one vein in this region were analyzed. *Figure 4b* compares the vessel diameters of these segments in anesthetized and awake states. For the selected artery, no significant change in diameter was observed, with both anesthetized and awake states showing an average diameter of 22 μm. In contrast, the selected vein displayed a notable response to isoflurane, with its diameter decreasing significantly from 54 μm under isoflurane anesthesia to 41 μm in the awake state. Flow velocity analysis of the same vessels further suggests that isoflurane may have a greater impact on venous blood flow velocity: arterial flow velocity increased slightly from 7.33 mm/s under anesthesia to 8.04 mm/s in the awake state, whereas venous flow velocity decreased from 10.48 to 8.11 mm/s, with both the magnitude and statistical significance of the decrease greater in the vein.

While single-vessel analysis provides valuable insights, its generalizability is limited. To validate the broader applicability of our findings, we conducted ROI-based analyses using fractional vessel area and mean velocity as primary metrics. These metrics extended the analysis of vessel diameter and flow velocity to entire brain regions or selected ROIs, which provides a more objective assessment of CBF changes at a global scale and reduces the bias associated with manually selecting vessel segments. For vessel area measurements, the term fractional denotes that the vessel area is normalized to the total area of the selected ROI. This normalization is essential for fair comparisons across ROIs of different sizes.

Our ROI-based analysis began with cortical arteries and veins. The relationship between arterio-venous classification and blood flow direction is more straightforward to infer on cortex, with downward flow representing cortical arteries and upward flow representing cortical veins (*Renaudin et al., 2022*). *Figure 4d* shows, for Mouse 1, the comparison of cortical upward (venous) and downward (arterial) flow density maps under different states. *Figure 3—figure supplement 1* also provides a detailed comparison of upward and downward flow for all three mice in both anesthetized and awake conditions. *Figure 4e* demonstrates that transitioning from isoflurane anesthesia to the awake state led to a reduction in vessel area for both arteries and veins within the selected cortical ROI across all three mice. However, the decrease in venous vessel area (averaging 55% across mice) was greater than that of arterial (averaging 35%). *Figure 4f* further illustrates the change in mean velocity within the selected ROI, where both arteries and veins showed a decreasing trend; however, venous flow velocity reduced by an average of 38%, compared to a 19% reduction in arterial flow velocity across the cortical ROI. These results underscore the different responses of arteries and veins to isoflurane anesthesia, which is also indicated by previous studies (*Sullender et al., 2022*; *Rakymzhan et al., 2021*; *Cao et al., 2017*).

To visualize flow velocity changes more clearly, we highlighted the flow velocity comparison for the same region in *Figure 4g* and provided a violin plot of velocity estimates within the ROI in *Figure 4h*. The violin plots reveal a consistently greater reduction in venous flow velocity from anesthesia to the awake state across all three mice.

For deeper brain regions, flow analyses were conducted without separating arteries and veins, as this physiological separation is not established for ULM in these regions. *Figure 4i* displays the selected ROIs for each mouse, while *Figure 4j, k* present the trends in vessel area and flow velocity

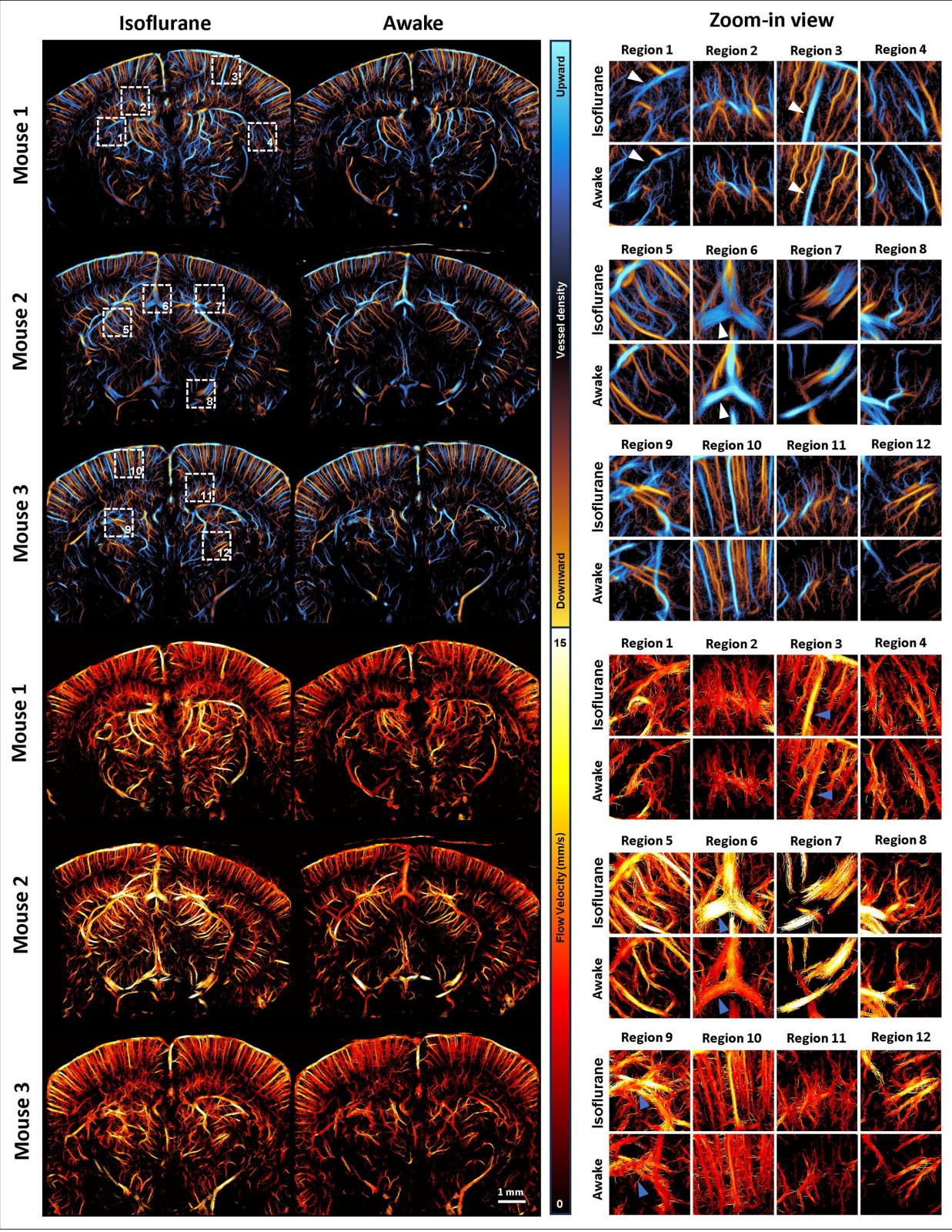

**Figure 3.** Comparison of ultrasound localization microscopy (ULM) images in isoflurane anesthetized and awake states. The ULM images of three different mice (Mouse 1–3) are shown. The upper panel shows the comparison of directional vessel density maps, while the lower panel shows the comparison of flow velocity. Four regions of interest (ROIs) are selected within each coronal plane (indicated by white dashed boxes in the whole-plane view of the vessel density map) for zoom-in comparison.

*Figure 3 continued on next page*

*Figure 3 continued*

The online version of this article includes the following figure supplement(s) for figure 3:

**Figure supplement 1.** Comparison of ultrasound localization microscopy (ULM) images in isoflurane anesthetized and awake states (Mouse 1–3), with distinctions made between upward and downward flow.

changes across different brain regions. It is worth noting that prior analyses (*Figure 4d–h*) aimed to illustrate arteriovenous differences. Since pial vessels are difficult to distinguish as arteries or veins based on flow direction in coronal plane imaging, they were excluded from the ROI selection in those analyses. In the current whole-brain comparisons (*Figure 4i–k*), the cortical ROIs no longer exclude pial vessels, since distinguishing between arteries and veins is not required. This aims to provide a more comprehensive representation of cortical vasculature. Results indicate that vessel area and flow velocity decreased across the brain as mice transitioned from anesthetized to awake states, with subcortical ROIs showing more pronounced changes than cortical ROIs. For example, in the thalamus and midbrain regions, fractional vessel area decreased by 37% on average and flow velocity decreased by 17%, whereas cortical ROIs exhibited smaller decreases of 4% in fractional vessel area and 11% in flow velocity. This work is consistent with a previous fMRI study, where it was also observed that isoflurane-induced cerebral hyperemia was not most pronounced in the cerebral cortex compared with other deeper brain regions (*Sicard et al., 2003*).

In summary, statistical analysis revealed a decrease in vessel area and blood flow velocity (particularly in venules) after awakening. These findings align with existing research, indicating higher blood perfusion during isoflurane anesthesia (*Sullender et al., 2022*; *Rakymzhan et al., 2021*; *Cao et al., 2017*; *Lyons et al., 2016*; *Takuwa et al., 2012*).

## Awake ULM imaging demonstrates high consistency in longitudinal imaging across different weeks

Longitudinal awake ULM brain imaging was feasible using the surgical and imaging techniques presented in this study. *Figure 5* presents the results of awake brain imaging performed on the same brain region over three consecutive weeks. Two ROIs at different depths were selected for each mouse to compare microvessel reconstruction across different time points. *Figure 5* demonstrates a high level of consistency in both directional vessel density maps and flow speed maps obtained from the three imaging sessions, although some minor discrepancies were observed. The inconsistency could potentially be attributed to physiological variation and/or slight misalignment.

To quantitatively evaluate the consistency of ULM imaging in longitudinal studies, we conducted a similar analysis to that shown in *Figure 4*. First, a case study was performed on individual vessels by selecting an artery and a vein in the cortex of Mouse 4 (*Figure 6a*) and measuring vessel diameter (*Figure 6b*) and flow velocity (*Figure 6c*) along the selected segments. The results show small variation in measurements across weeks. For vessel diameter, both arterial and venous segments demonstrated statistical equivalence across the three weekly measurements. Statistical testing of equivalence was conducted using the two one-sided test (TOST) procedure, which evaluates whether the difference between two time points falls within a predefined equivalence margin. Specifically, equivalence is defined as the inter-week difference being smaller than three times the standard deviation of 1 week. A statistically significant result in TOST (p < 0.001) supports the interpretation that the measurements are statistically equivalent, which is denoted as 'equiv.' in the figures. For flow velocity, apart from a slight decrease in the venous flow during the second week (4.49 mm/s, compared to 6.03 mm/s in the first week and 6.00 mm/s in the third week), no other significant differences were observed. Overall, quantitative measurements maintained high consistency, though slight variations were expected due to differences in probe positioning and physiological variations among the mice.

To further generalize these findings and examine longitudinal variation in ROI-based analysis, we used Mouse 4 as an example to show the consistency of blood flow density across different flow directions in the cortex (*Figure 6d*) and extended the quantitative analysis to all three mice (*Figure 6e*) (individual ULM upward and downward flow images for all three mice over the 3-week longitudinal study period can be found in *Figure 5—figure supplement 1*). In the comparative study shown in *Figure 4*, clear differences were seen between arterial and venous responses under various conditions, while in the longitudinal study, both arteries and veins exhibited similar levels of variation. For

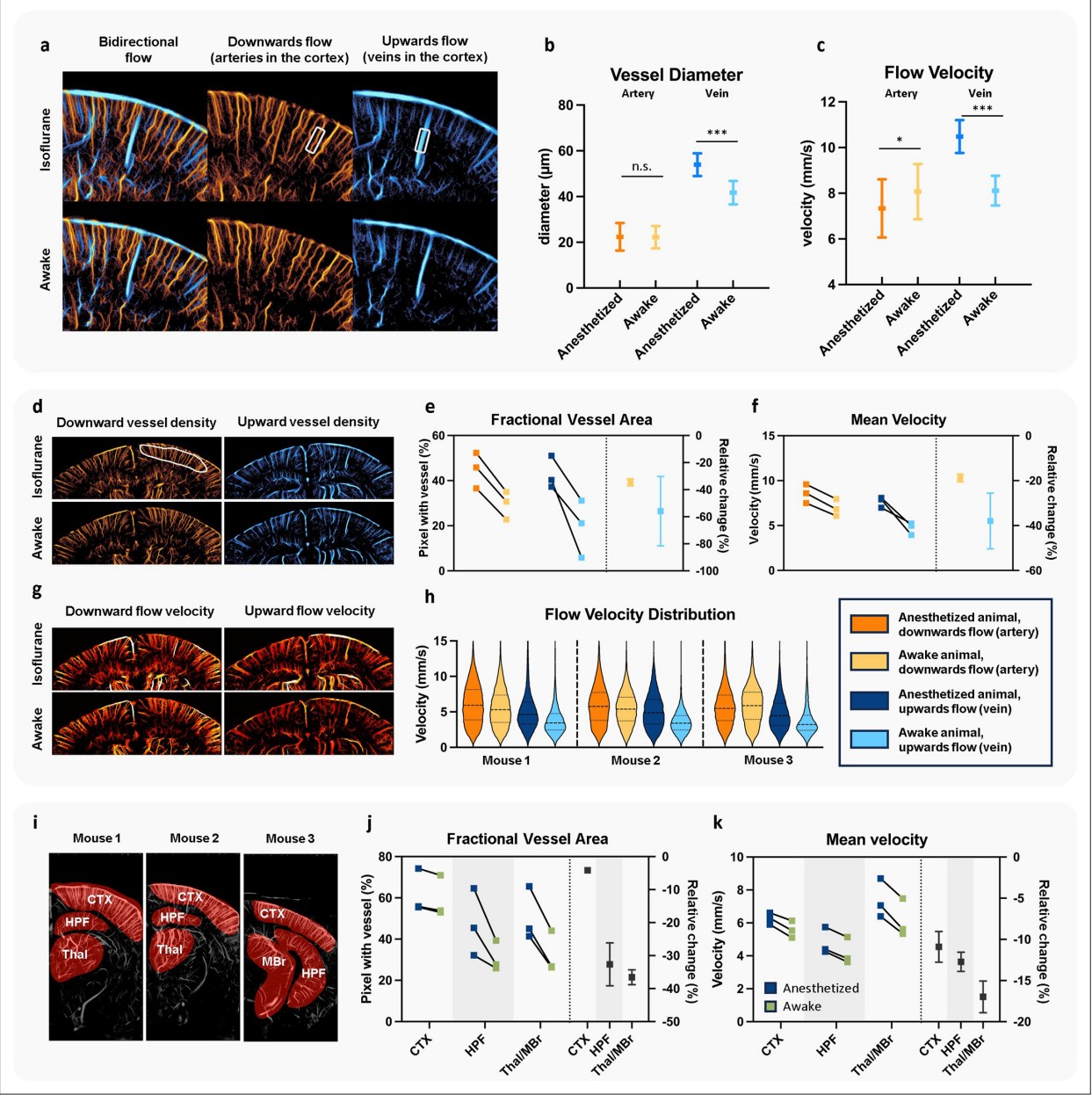

**Figure 4.** Quantitative comparison of the ultrasound localization microscopy (ULM) images in isoflurane anesthetized and awake states. (**a**) Example cortical region near region 3 in Mouse 1 (from **Figure 3**), with vessel density maps displayed separately by flow direction. In this cortical region, upward flow corresponds to venous blood, while downward flow corresponds to arterial blood. One artery and one vein segment were selected within the boxed areas for analysis of flow velocity and diameter. Comparisons of vessel diameter (**b**) and flow velocity (**c**) for the selected arterial and venous segments. Statistical analysis was conducted using *t*-test at each measurement point along the segments. (**d**) Whole cortical region in Mouse 1, showing separate blood flow density maps for upward (venous) and downward (arterial) flow, with a defined region of interest (ROI, white circled region) used for quantitative analysis. Quantitative assessment of fractional vessel area (**e**) and mean flow velocity (**f**) within the cortical ROI for Mouse 1–3, presented as individual values in each state (left *y*-axis) and the relative reduction from the anesthetized baseline to the awake state (right *y*-axis). (**g**) Flow velocity map for the same cortical region shown in (**d**) using Mouse 1 as an example. (**h**) Flow velocity distributions within the cortical ROI for Mouse 1–3, comparing anesthetized and awake states. (**i**) Regional analysis across different brain areas using bidirectional ULM maps, with selected ROIs from the cortex (CTX), hippocampal formation (HPF), thalamus (Thal), and midbrain (MBr) for each mouse. Quantitative analysis of fractional vessel area (**j**) and mean blood flow velocity (**k**) for the ROIs in each brain region in Mouse 1–3. Colorbars of (**a, d, g**) are the same as in **Figure 3** (*p < 0.05, ***p < 0.001).

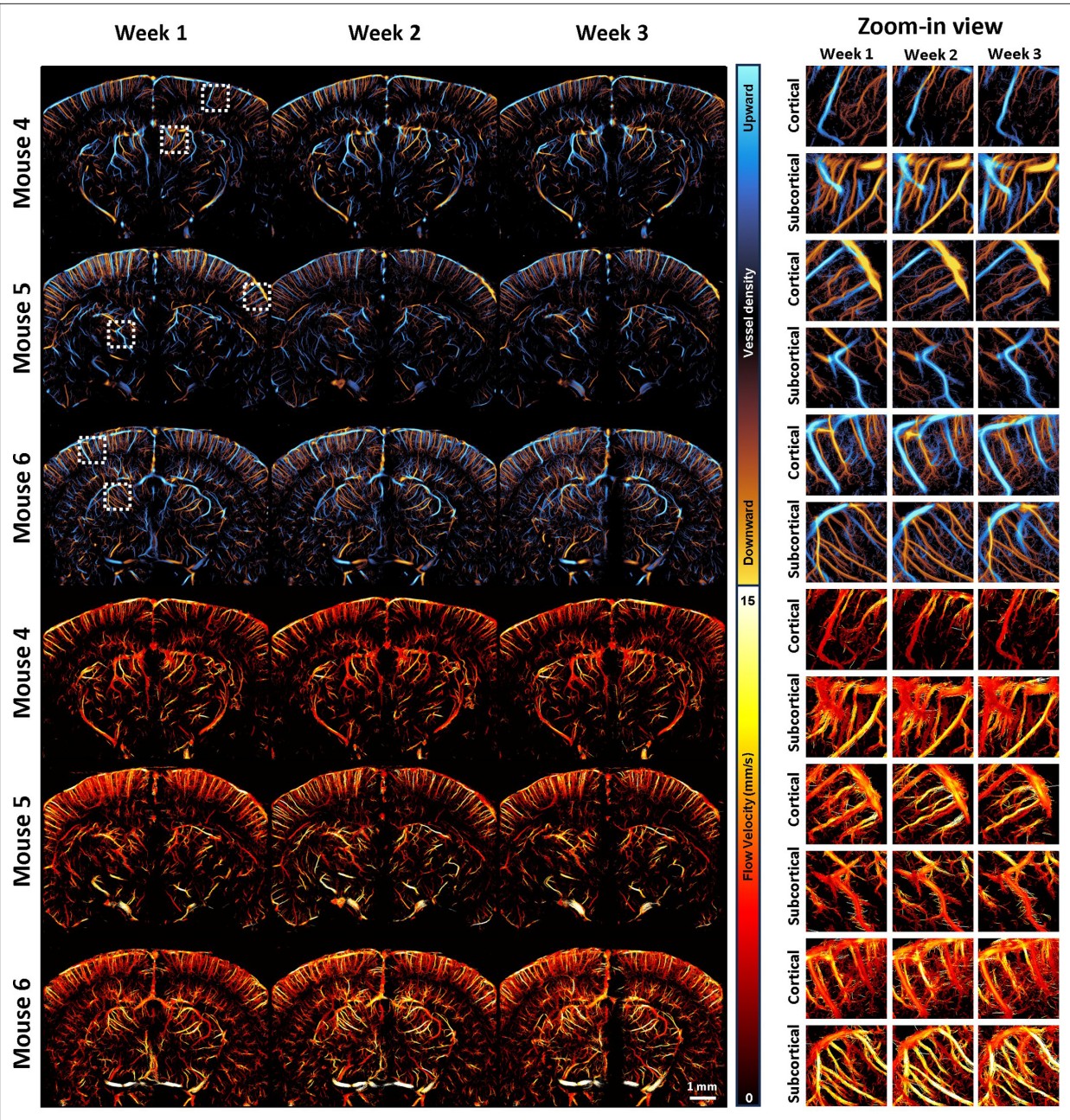

**Figure 5.** Longitudinal awake ultrasound localization microscopy (ULM) imaging results on the same coronal plane for three consecutive weeks. The ULM images of three different mice (Mouse 4–6) are shown. The upper panel shows the comparison of directional vessel density maps, while the lower panel shows the comparison of flow velocity. Two regions of interest (a cortical ROI and a subcortical ROI) are selected within each coronal plane (indicated by white dashed boxes in the whole-plane view of the vessel density map) for zoom-in comparison.

The online version of this article includes the following figure supplement(s) for figure 5:

**Figure supplement 1.** Longitudinal comparison of Mouse 4–6 with distinctions made between upward and downward flow.

instance, in Mouse 6, both arterial and venous fractional vessel area showed a notable decrease from around 60% in the first week to around 40% in the second week, which is a relatively large variation compared with other measurements. This quantitative analysis aims to inform readers of the expected range of variation of this technique. *Figure 6f* displays the flow velocity map for the same region as in *Figure 6d*, with flow velocity distributions inside the selected ROI for all three mice shown in *Figure 6g*. While arterial and venous flow velocity distributions exhibit clear distinctions, the distribution shapes remained relatively consistent across the 3 weeks. Specifically, variation in median velocity

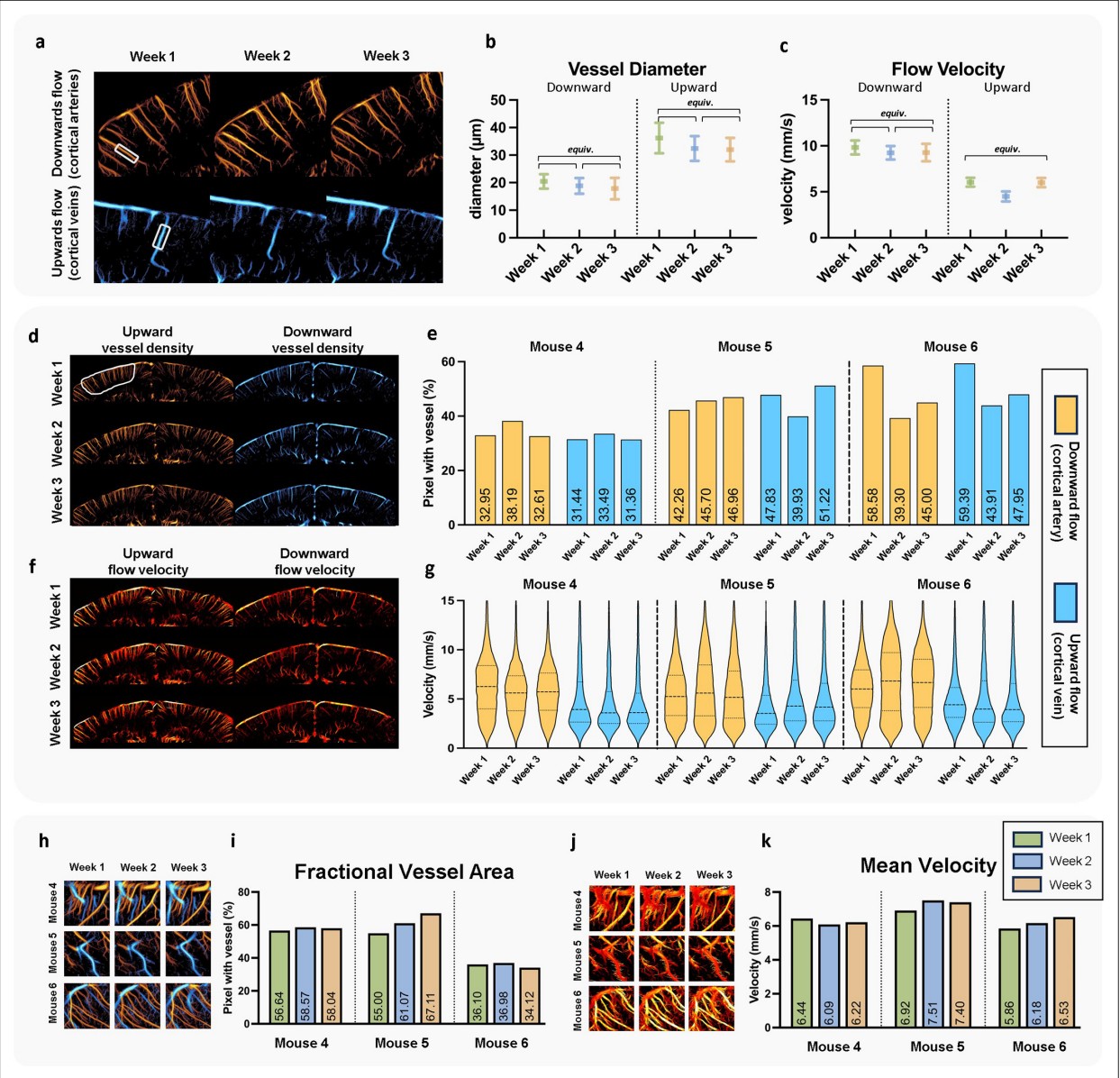

**Figure 6.** Analysis of ultrasound localization microscopy (ULM) images acquired in the three consecutive weeks. (**a**) Two example cortical regions in Mouse 4 (from ***Figure 5***), with one for artery selection and the other for vein. The vessel segments were selected within the boxed areas for analysis of flow velocity and diameter. Comparisons of vessel diameter (**b**) and flow velocity (**c**) for the selected arterial and venous segments. Statistical analysis was performed using the two one-sided test (TOST) to evaluate consistency of measurement. The label 'equiv.' indicates statistically equivalent measurements (p < 0.001), defined as inter-week differences smaller than three times the standard deviation of 1 week. (**d**) Whole cortical region in Mouse 4, showing separate blood flow density maps for upward (venous) and downward (arterial) flow, with a defined region of interest (ROI, white circled region) used for quantitative analysis. (**e**) Bar plot for the fractional vessel area comparison of the cortical artery and vein among Mouse 4–6. (**f**) Flow velocity map for the same cortical region shown in (**d**) using Mouse 4 as an example. (**g**) Flow velocity distributions within the cortical ROI for Mouse 4–6, comparing arterial and venous flow across 3 weeks. (**h**) Bidirectional ULM vessel density map of subcortical ROIs for the three different mice. (**i**) Bar plot for the fractional vessel area comparison of the subcortical ROIs in (**h**). (**j**) ULM flow velocity map of the same subcortical ROIs for the three different mice. (**k**) Bar plot for the mean velocity comparison of the subcortical ROIs.

was within 1 mm/s. In contrast, anesthesia-induced changes can lead to velocity shifts exceeding 1 mm/s. Even in the previously mentioned case of Mouse 6 during weeks 1 and 2, arterial mean flow velocity only shifted from 6.31 mm/s in the first week to 7.00 mm/s in the second week.

We also conducted ROI analyses without distinguishing between arteries and veins for subcortical regions (***Figure 6h–k***), using the same ROIs as those shown in ***Figure 5***. Despite subcortical regions showing the largest vessel area variability in anesthesia-induced changes, vessel area in those regions

was relatively stable in the longitudinal study. In terms of vessel area (*Figure 6i*), Mouse 5 exhibited the greatest variation among the three mice, with vessel area increasing from 55% in the first week to 67% in the third week. Compared to the maximum 40% reduction seen from anesthesia to awake states in *Figure 4j*, vessel area measurement in subcortical regions remained consistent. This observation also applied to mean velocity measurements.

## Discussion

In this study, we introduced a method for performing ULM brain imaging in awake mice under a longitudinal study setting. Our method enabled high-resolution imaging of deep cerebral microvasculature with the animal under the awake state. We translated the awake imaging techniques previously described in fUS (*Brunner et al., 2021*) to our study to enable awake ULM and established a quantitative metric for ULM image reconstruction. Based on the setup above, we studied CBF changes induced by anesthesia, which aligned well with literature. Isoflurane has been shown to increase vascular diameter and CBF in mice, as validated by multiple imaging modalities including optical coherence tomography (*Rakymzhan et al., 2021*), photoacoustic microscopy (*Cao et al., 2017*), two-photon microscopy (*Lyons et al., 2016*), and laser speckle imaging (*Sullender et al., 2022*; *Takuwa et al., 2012*). These effects have also been validated in larger animal models such as rats (*Sicard et al., 2003*), dogs (*Iida et al., 1998*), and marmosets (*Santisakultarm et al., 2016*). In humans, vasodilation and increased CBF caused by volatile anesthetics such as isoflurane have also been reported (*Slupe and Kirsch, 2018*).

Statistical analysis from *Figure 4* shows that certain vessels exhibit a larger diameter under isoflurane anesthesia, and the fractional vessel area, calculated as the percentage of vascular area within selected brain region ROIs, is also higher in the anesthetized state. These findings suggest a vasodilation effect induced by isoflurane, consistent with existing research (*Sullender et al., 2022*; *Rakymzhan et al., 2021*; *Cao et al., 2017*; *Lyons et al., 2016*; *Takuwa et al., 2012*). It is worth noting that although our data indicate a global elevation of CBF under isoflurane anesthesia, individual vessels exhibit large discrepancies in behavior. For example, the vessel at the left lower corner from Mouse 3 in the entorhinal cortex (*Figure 3*) shows almost no blood flow during anesthesia but then exhibits high vessel pixel density after awakening. The wide range of vessel behaviors was also previously reported in literature (*Sullender et al., 2022*; *Santisakultarm et al., 2016*). Our results indicate that awake ULM imaging has ample spatial resolution and imaging depth of penetration to resolve individual vessel variations down to micron-sized vessels deep into the brain. This is a unique capability that is not available from other biomedical imaging modalities.

Increased blood flow velocity induced by isoflurane has also been reported by other studies (*Rakymzhan et al., 2021*; *Cao et al., 2017*). However, previous research presented different speculations on the predominant factor contributing to the increase in CBF induced by anesthesia, specifically whether the increase is attributed to vasodilation or increase in blood flow velocity (*Sullender et al., 2022*). One study found significant changes in both blood flow and vessel diameter but minor changes in flow velocity, suggesting that the increase in blood flow was largely driven by vasodilation (*Rakymzhan et al., 2021*). Conversely, another study drew the opposite conclusion (*Sullender et al., 2022*). Benefiting from the large field of view of ULM and its capability to directly quantify microvascular blood flow velocity, we can make a more comprehensive inference regarding the relationship among the changes in vessel diameter, flow speed, and flow volume from anesthetized to awake states. For arteries, the change in blood flow velocity is not significant, indicating that the alteration in blood flow may be primarily due to vasodilation instead of velocity change. Isoflurane causes vasodilation by acting on the ion channels (e.g., potassium channel) of smooth muscle (*Iida et al., 1998*), which is more abundantly found in arteries than in veins. In the case of veins, which do not actively dilate or constrict, their vessel diameter and blood flow variations are more likely controlled by passive mechanisms. *Figure 5* reveals significant differences in flow velocity of veins between anesthesia and awake state, suggesting that the changes in flow velocity may have a greater impact on venous blood flow volume compared with arterial volume.

The differences in cerebral vasculature between anesthetized and awake states observed using ULM are also in agreement with other studies (*Sullender et al., 2022*; *Rakymzhan et al., 2021*; *Cao et al., 2017*; *Takuwa et al., 2012*). However, previous studies mostly used optical imaging techniques, which have limited penetration depth and can only observe surface pial vessels in the cortex. Some

other studies using fMRI can detect deeper CBF changes in the whole brain, but they do not provide insights about small vessel blood flow variations due to insufficient spatial resolution (*Sicard et al., 2003*). As a bridging imaging modality between MRI and optical techniques, awake ULM enables observations of detailed microvascular variations induced by anesthesia across the whole depth of the brain, which provides complementary information to existing biomedical imaging modalities.

Although isoflurane is widely used in ultrasound imaging because it provides long-lasting and stable anesthetic effects, it is important to note that the vasodilation observed with isoflurane is not representative of all anesthetics. Some anesthesia protocols, such as ketamine combined with medetomidine, do not produce significant vasodilation and are therefore preferred in experiments where vascular stability is essential, such as functional ultrasound imaging (*Réaux-Le-Goazigo et al., 2022*). Our current study primarily focused on demonstrating the feasibility of longitudinal ULM imaging in awake animals, instead of conducting a systematic investigation of how isoflurane anesthesia alters CBF. Due to the limited number of animals used, the analyses presented in this work should be interpreted as example case studies. While the trends observed across animals were consistent, the small sample size restricts the scope of statistical inference. For future work, it would be valuable to design more rigorous control experiments with larger sample sizes to systematically compare the effects of isoflurane anesthesia, awake states, and other anesthetics that do not induce vasodilation on CBF.

Our proposed method enabled repeatable longitudinal brain imaging over a 3-week period, addressing a key limitation of conventional ULM imaging and offering potential for various preclinical applications. However, there are still some limitations in this study.

One of the limitations is the lack of objective measures to assess the effectiveness of head-fix habituation in reducing anxiety. This may introduce variability in stress levels among mice. Recent studies suggest that tracking physiological parameters such as heart rate, respiratory rate, and corticosterone levels during habituation can confirm that mice reach a low stress state prior to imaging (*Chabouh et al., 2024*). This approach would be highly beneficial for future awake imaging studies. Furthermore, alternative head-fixation setups, such as air-floated balls or treadmills, which allow the free movement of limbs, have been shown to reduce anxiety and facilitate natural behaviors during imaging (*Bertolo et al., 2021*). Adopting these approaches in future studies could enhance the reliability of awake imaging data by minimizing stress-related confounds.

Another limitation of this study is the potential residual vasodilatory effect of isoflurane anesthesia on awake imaging sessions and the short imaging window available after bolus injection. The awake imaging sessions were conducted shortly after the mice had emerged from isoflurane anesthesia, required for the MB bolus injections. The lasting vasodilatory effects of isoflurane may have influenced vascular responses, potentially contributing to an underestimation of differences in vascular dynamics between anesthetized and awake states. In addition, since MBs are rapidly cleared from circulation, the duration of effective imaging is limited to only a few minutes, which also overlaps with the anesthesia recovery period, constraining the usable awake-state imaging window. Future improvement on MB infusion using an indwelling jugular vein catheter presents a promising alternative to address these limitations. This method allows for stable MB infusion without the need for anesthesia induction, ensuring that the awake imaging condition is free from residual anesthetic effects. Moreover, it has the potential to extend the duration of imaging sessions, offering a longer and more stable time window for data acquisition. Furthermore, by performing ULM imaging in the awake state first, instead of starting with anesthetized imaging, researchers can achieve a more rigorous comparison of how various anesthetics influence cerebral microvascular dynamics relative to the awake baseline.

In our longitudinal study, consistent imaging results were obtained over a 3-week period, demonstrating the feasibility of awake ULM imaging for this duration. However, for certain research applications, a monitoring period of several months would be valuable. Extending the duration of longitudinal awake ULM imaging to enable such long-term studies is a potential direction for future development.

Tissue motion is also a critical concern of ULM imaging. While rigid motion correction is often effective in anesthetized animals, awake animal imaging presents greater challenges due to the more prominent non-rigid motion, particularly in deeper brain regions. This is evidenced in *Figure 1— figure supplement 1* (Mouse 7), where cortical vessels remain relatively stable, but regions around the colliculi and mesencephalon exhibit more noticeable motion artifacts, indicating that displacement is more pronounced in deeper areas. To address these deeper, non-rigid motions, recent studies suggest estimating non-rigid transformations from unfiltered tissue signals before applying corrections to ULM

vascular images (*Renaudin et al., 2022*; *Hingot et al., 2017*). Such advanced motion correction strategies may be more effective for awake ULM imaging, which experiences higher motion variability. The development of more robust and effective motion correction techniques will be crucial to reduce motion artifacts in future awake ULM applications. Meanwhile, with 2D imaging, we cannot correct for out-of-plane motion, which necessitates 3D imaging. In the future, 3D motion correction techniques that account for complex tissue motions and are computationally efficient need to be developed for awake and longitudinal ULM imaging.

Advances in ULM imaging methods can benefit longitudinal awake imaging. For instance, dynamic ULM can differentiate between arteries and veins by leveraging pulsatility features (*Bourquin et al., 2022*). 3D ULM, with volumetric imaging array (*McCall et al., 2023*; *Heiles et al., 2019*), enables the reconstruction of whole-brain vascular network, providing a more comprehensive understanding of vessel branching patterns. Meanwhile, 3D ULM also helps to mitigate the challenge of aligning the identical coronal plane for longitudinal imaging, a process that requires precise manual alignment in 2D ULM to ensure consistency. Additionally, this alignment issue can also be alleviated in 2D imaging using backscattering amplitude method, which may assist in estimating out-of-plane positioning during longitudinal imaging (*Renaudin et al., 2023*).

Longitudinal brain imaging in the awake state offers a promising tool for neuroscience research as it not only avoids the confounding effects of anesthesia on cerebral vasculature, but also enables observations of intrinsic dynamics of the vasculature within the same subject, minimizing potential sources of bias associated with inter-subject variability. In the future, this technique is expected to be further integrated with disease models to study the changes in cerebral vasculature during the development of diseases. Also, this technique can be further combined with the latest functional ULM (fULM) studies (*Renaudin et al., 2022*) to allow awake fULM imaging. Our study laid the foundation for these studies with awake fULM, which is expected to improve the sensitivity of conventional fULM techniques because hemodynamic responses are much stronger in the awake state than in anesthesia (*Aksenov et al., 2015*; *Pisauro et al., 2013*; *Desai et al., 2011*). However, it is also important to note that although longitudinal awake imaging presents promise to avoid the confounding effects of anesthetics, imaging under anesthesia remains more convenient and controllable in many cases. For applications where the physiological question of interest is not sensitive to anesthesia-induced vascular effects, anesthetized imaging still offers a simpler and more stable approach. Awake imaging inherently exhibits greater physiological variability. However, care must be taken at the experimental level to minimize confounding sources of variation, such as stress level of the animal or handling inconsistencies, to ensure that the measurements are physiologically meaningful.

## Methods
### Animal preparation and usage overview
Eight healthy female C57BL/6J mice (8–12 weeks) (from The Jackson Laboratory) were used for this study, numbered as Mouse 1–8. Three mice (Mouse 1–3) were used to compare imaging results between awake and anesthetized states (*Figures 3 and 4*). Three additional mice (Mouse 4–6) underwent longitudinal imaging over a 3-week period (*Figures 5 and 6*). Among them, Mouse 4 was also used as an example to demonstrate the overall system schematic and saturation conditions (*Figures 1 and 2*). Several mice (Mouse 2, 6, 7, and 8) exhibited suboptimal cranial window quality or image artifacts and were included to illustrate common surgical or imaging issues (*Figure 1—figure supplement 1*). The specific usage of each animal is also annotated in the corresponding figure captions. All experimental procedures were conducted in accordance with the guidelines set by the University of Illinois Institutional Animal Care and Use Committee (IACUC Protocol number #22165). The animals were housed in an environment with a 12-hr light/dark cycle and had free access to food and water. Prior to the cranial window surgery, the mice were kept in group housing, and they were individually housed post-surgery.

### Pre-surgery handling
A week before performing the cranial window surgery, the animals underwent tunnel handling, a procedure shown to significantly reduce the levels of anxiety in mice (*Hurst and West, 2010*; *Gouveia and Hurst, 2017*). Specifically, a commercially available polycarbonate mouse transfer tube (TRANS-TUBE

130X50MM, Braintree Scientific, Inc) was used for the initial 3 days. The mice were encouraged to enter the tube from their cage, after which the tube was lifted, allowing the mouse to remain inside for 30 s. The mouse was then returned to its cage for free movement for 1 min, before a second identical tunnel handling was carried out. This procedure was done twice daily. Approximately 3 days into this routine, the mice were accustomed to the tunnel handling method. The animals were then picked up using the commercial tunnel, and then a 3D-printed body tube was attached to the other end of the tunnel. The mice were allowed to enter the body tube voluntarily. Subsequently, the body tube was utilized for handling the mice in the following 5 days as a replacement for the commercial tunnel.

## Cranial window surgery

The entire procedure closely followed a previously published protocol (*Brunner et al., 2021*). To minimize brain swelling, the animal received an intraperitoneal injection of Dexamethasone (0.5 mg/kg body weight). The mouse was anesthetized by inhalation of isoflurane (3% for induction and 1–1.5% for maintenance). After confirming that the mouse was anesthetized, the head of the animal was fixed in the stereotaxic frame. Afterward, the scalp of the mouse was incised, and the temporalis muscle was dissected from the skull using forceps. Tissue adhesive (3M, cat. no. 70200742529) was applied to secure the retracted muscles and incision edges, ensuring proper closure of the skin. Subsequently, a headpost was fixed to the skull using tissue adhesive and dental cement (Sun Medical Co, Ltd, cat. no. Super-Bond C&B). The skull was then carefully thinned using a high-speed rotary micromotor (Foredom, cat. no. K.1070) to create a cranial window with a width of approximately 8 mm laterally and which extends from bregma to lambda in the anterior–posterior direction. The skull was continuously thinned along this outline until it became loose, allowing for separation of the bone from the dura mater using forceps. After removing the skull, a PMP plastic sheet (Goodfellow, cat. no. ME311051) was placed over the cranial window and fixed to the bone using tissue adhesive and dental cement. To provide protection, the window was covered with biocompatible silicone rubber (Smooth-on, cat. no. Body Double-Fast Set). Once the silicone rubber was secure, the anesthesia was discontinued, and the mouse was allowed to regain consciousness.

## Post-surgical care and head-fix habituation

Following the surgery, subcutaneous administration of Carprofen (5–10 mg/kg body weight) was provided to the animals for immediate post-operative pain management. The animals were allowed to recover from anesthesia in their individual cages, with a heating pad provided to maintain optimal body temperature during the recovery period. Observations were made every 15 min until the animals reached sternal recumbency. To aid in their recovery, dry food soaked in water was provided, along with the use of recovery gel (ClearH2O, cat. no. 72-06-5022) to facilitate chewing and hydration. In the case that any signs of pain were observed, additional subcutaneous doses of 5–10 mg/kg Carprofen were administered every 12–24 hr to alleviate surgical discomfort for 3 days. Post-operative monitoring of the animals was performed daily for a duration of 14 days following the surgery. Once the animals had fully recovered from the surgical procedure, the animals were habituated daily to head fixation after walking through the body tube. The duration of the head-fixation periods was gradually increased over time, starting at approximately 10 min on day 1 and extending to up to 1 hr after 3 days (*Brunner et al., 2021*). If any signs of discomfort, such as excessive movement or vocalization, were observed during the head-fixation sessions, then the sessions would be immediately discontinued.

## The use of 3D-printed body tube

The body tube was secured to the table using four screws on the outer side, while two small screws on the top were used to clamp the headpost. Two semicircular grooves that fit the size of the small screws were made on the headpost (*Figure 1a*). During the process of head-fixing the animal, the experimenter guided the animal to enter the body tube from the rear. Once the animal protruded its head from the front end, the experimenter manually grasped the headpost, gently restrained the head of the animal, and subsequently aligned the semicircular grooves with the screws to immobilize the animal.

## Experimental procedure of imaging sessions

On the day of imaging, the animal was guided to walk into the body tube, and the headpost was firmly secured to the tube. The protective silicone rubber and headpost were cleaned using 70% ethanol, followed by rinsing with sterile saline. Then, forceps were inserted between the silicone rubber and the cement to detach the silicone rubber from the headpost. Due to the challenges of conducting continuous infusion in awake animals, contrast MBs were administered via tail vein bolus injection in this study. Animals were anesthetized with isoflurane (3% for induction and 1% for maintenance) before tail vein catheterization to alleviate pain and stress. Ultrasound coupling gel was applied, and the transducer was positioned to find the imaging plane. When the imaging plane was identified, the relative position of the probe and the headpost was recorded, serving as a reference point for subsequent longitudinal studies. B-mode and power Doppler images were also saved as references to facilitate the identification of the same imaging plane in the following weeks.

Once the imaging plane was confirmed, the isoflurane anesthesia was terminated, and 0.1 ml of DEFINITY (Lantheus, North Billerica, MA) MB was administered via the tail vein catheter. The completion of the injection was considered as the starting point, denoted as $T = 0$. From $T = 0$ onwards, continuous ultrasound data was saved. After completion of the imaging session, the cranial window was filled with the same biocompatible silicone rubber described above, allowing approximately 10 min for solidification. The headpost was then removed from the body tube, and the animal was returned to its home cage. Imaging sessions were conducted once per week over a 3-week period.

For the experiment of comparing differences between anesthesia and awake state (*Figures 3–5*), once the mice were fixed to the body tube, they were continuously anesthetized with 2% isoflurane for more than 15 min. During this process, the tail vein catheterization was completed, and the imaging plane was confirmed. Following the bolus injection of 0.1 ml MB, ULM data under anesthesia was acquired for 1000 s. The tail vein catheter remained in place. After the ULM acquisition under anesthesia, it was essential to wait until no bubbles were circulating within the blood stream before terminating anesthesia. Once the animal approached full arousal, another bolus injection of 0.1 ml MB was administered. The catheter was then removed, and ULM acquisition started (marked as $T = 0$). Other procedures were the same as those described in the previous two paragraphs. For the ULM data collected under anesthesia, since there was no specific time point at which mice achieved full pupillary dilatation, we consistently employed the radiofrequency (RF) dataset starting from $T = 500$ s until the cumulative MB count reached 5 million for ULM reconstruction to make the MB concentration similar to that in the awake cases.

## Pupillary recording and measurement

Monitoring the pupillary area of rodents is commonly used to determine their level of arousal (*Privitera et al., 2020*). The pupils constrict under isoflurane anesthesia and enlarge upon awakening (*Turner et al., 2023*; *Kum et al., 2016*). In this study, video recording of pupillary area was performed using a camera (ace acA800-510um, Basler Inc, Exton, PA) placed in front of the mouse, with timestamps synchronized to the ultrasound acquisition. The ULM imaging session was carried out in a dark room. An IR flashlight (EVOLVA Future Technology T20) was positioned above the camera. Since mice are functionally blind to IR wavelengths (*Breuninger et al., 2011*; *Chang et al., 2013*), the use of IR illumination can minimize the effect of light exposure on pupil changes so that the pupillary area is mostly influenced by anesthesia. The videos recorded by the camera were analyzed using ImageJ software, and the contour of the pupil was manually circled. The equivalent area of a single pixel was calibrated, which allowed for the quantification of the pupillary area according to the number of pixels occupied by the pupil.

## Ultrasound imaging sequence

A Vantage 256 system (Verasonics Inc, Kirkland, WA) was used for this experiment. The ultrasound system was connected to a linear-array transducer (L35-16vX, Verasonics Inc), which was fixed onto a 3D-printed holder attached to a translation motor (VT-80, Physik Instrumente, Auburn, MA). The motor allowed precise control of transducer movement in the elevational direction, enabling adjustment of the imaged coronal plane. Ultrasound was transmitted at a center frequency of 20 MHz. A 9-angle compounded plane wave technique was used (angles: –8°, –6°, –4°, –2°, 0°, 2°, 4°, 6°, and 8°), with a post-compounded frame rate of 1000 Hz. Interleaved sampling is employed to capture high-frequency

echoes more effectively. With the system's sampling rate limited to 62.5 MHz, the upper limit of the center frequency of the transducer passband is 15.625 MHz. To mitigate aliasing, two transmissions are sent per angle, staggered in time. This approach effectively doubles the sampling rate, ensuring more accurate image reconstruction. The ensemble size of each dataset was 800 frames, and the acquired RF data was saved for offline reconstruction. Beamforming was conducted using the Verasonics built-in program to reconstruct the in-phase and quadrature (IQ) data. The IQ data had a pixel size of half the wavelength in the axial direction and one wavelength in the lateral direction.

## ULM image processing

To ensure consistent localization of bubbles at different depths, a noise equalization method was employed to adjust signal intensity (*Song et al., 2017a*). High-pass filtering with a cutoff frequency of 30 Hz was applied to the IQ data to enhance sensitivity for MB extraction. MBs moving toward and away from the transducer were separated into two distinct datasets (upward flow and downward flow) based on the positive and negative Doppler shifts (*Huang et al., 2020*). Subsequently, singular value decomposition filtering was applied to the IQ data to further eliminate clutter signals from tissue and extract MB signals (*Demené et al., 2015*; *Baranger et al., 2018*). The singular value cutoff was adaptively determined to achieve objective and stable filtering results (*Song et al., 2017b*). The filtered MB data was then spline interpolated to a resolution with a lateral and axial pixel size of one-tenth of the wavelength (4.928 µm). Each interpolated frame was subjected to 2D normalized cross-correlation with the point spread function of the imaging system, which was empirically determined. Regional maxima in the cross-correlation results indicated the center position of MBs (*Song et al., 2018*). Thresholding of image intensity was applied to reject low-intensity values and prevent noise from being erroneously identified as bubbles. The coordinates of the MB center points obtained from each frame were tracked using the uTrack algorithm (*Jaqaman et al., 2008*; *Stein and Thiart, 2016*). To ensure the detection of reliable MB tracks, only tracks with a minimum length of 10 consecutive frames were considered valid. The choice of 10 consecutive frames (10 ms) was based on established practice (*Errico et al., 2015*) but can be adjusted as needed (*Couture et al., 2018*). For the uTrack algorithm, two additional key parameters were specified: the maximum linking distance and the gap-filling distance, both set to 10 pixels (~50 µm). This configuration means that only bubble centroids within 10 pixels of each other across consecutive frames are considered part of the same bubble trajectory. Additionally, when the start and end points of two tracks fall within this threshold, the gap-filling parameter merges them into a single, continuous track. It is important to select these parameters carefully, as overly large values could lead to an overestimation of flow velocity. By setting the maximum linking distance to 10 pixels, we effectively limited the measurable velocity to 50 mm/s, under the assumption that no bubble would exceed a 50-µm displacement within the 1-ms interval between frames. Considering the velocity distribution across the mouse brain (*Demeulenaere et al., 2022*), this 50 mm/s limit ensures that most of the blood flow is captured accurately. After determining bubble tracks with the specified parameters for the uTrack algorithm, accumulating the MB tracks resulted in the flow intensity map. The final values were square-root transformed to compress the dynamic range of the image for display. Furthermore, the distance traveled by MB between adjacent frames was calculated during the tracking process to determine blood flow velocity, which was then assigned to each individual bubble track. The average velocity computed from multiple tracks was used to generate a comprehensive flow speed map.

## Quantitative analysis of ULM images

For single-vessel analysis, the vessel density map (either bidirectional or unidirectional) was binarized, and the center line of the vessel was extracted using MATLAB 'bwskel' function. The radius of each vessel was defined as the distance from the central line to the nearest vessel edge on the binarized map, with diameter calculated by doubling this radius. Given a pixel size of approximately 4.928 µm, the minimum detectable vessel diameter using this approach is 9.856 µm, explaining why the smallest measured radius in unsaturated conditions, as shown in *Figure 2b*, always falls into this value. To analyze a specific vessel, we manually selected a vessel segment. Each point on the central line provided a diameter measurement. For flow velocity, each non-zero pixel within the chosen segment was used to estimate effective blood flow velocity in the vessel.

For ROI-based analysis, ROIs were manually defined based on a brain atlas to ensure consistency across both anesthetized and awake states, as well as across weekly imaging sessions, enhancing comparability of results. In ROI-based analysis, we focused on two primary parameters: fractional vessel area and mean velocity. Fractional vessel area was defined as the proportion of the pixel count occupied by blood vessels within each ROI, obtained by binarizing the ULM vessel density maps and calculating the percentage of the pixels with MB signal. Mean velocity was calculated by averaging all non-zero pixel of velocity estimates within the ROI. The velocity distribution within each ROI was also visualized using violin plots, as shown in *Figures 2, 4, and 6*, to illustrate the range and density of flow velocity estimates across different acquisition. In this study, we focused on these two metrics because they represent the most straightforward extension of single-vessel analysis to brain-wide vascular changes.

## Code availability

To support quantitative longitudinal analysis of ULM data, we developed an open-source MATLAB application (https://github.com/ekerwang/ULMQuantitativeAnalysis copy archived at *Wang, 2025*). This tool is designed to facilitate ROI-based analysis of ULM images for longitudinal comparisons. It supports multiple quantification metrics, including but not limited to vessel area and mean velocity used in this study. Users can select and adapt different metrics based on their specific applications, as a wide range of ULM-based quantification metrics have been developed for different pathological and pharmacological studies.

## Acknowledgements

This study was partially supported by the National Institute on Aging, the National Institute of Biomedical Imaging and Bioengineering, and the National Institute of Neurological Disorders and Stroke of the National Institutes of Health under grant numbers R21EB030072, R21EB030072-01S1, R21AG077173, R56NS131516, and by the National Science Foundation CAREER Award 2237166, and by the Chan Zuckerberg Initiative (CZI) Ben Berres Early Career Award. The content is solely the responsibility of the authors and does not necessarily represent the official views of the NIH and NSF. MRL was supported by a Beckman Institute Postdoctoral Fellowship. We thank Dr. Baher Ibrahim and Mr. Gang Xiao from Dr. Daniel Llano's lab for their assistance with the awake imaging setup. We thank Dr. Danqing Hu from Emory University for her insights regarding the impact of anesthetics on cerebral blood flow.

## Additional information

### Funding

| Funder | Grant reference number | Author |
| --- | --- | --- |
| National Institutes of Health | R56NS131516 | Yike Wang<br>Matthew R Lowerison<br>Qi You<br>Bing-Ze Lin<br>Daniel A Llano<br>Pengfei Song |
| National Science Foundation | CAREER Award 2237166 | Pengfei Song |
| Chan Zuckerberg Initiative | Ben Berres Early Career Award | Matthew R Lowerison<br>Zhe Huang<br>Pengfei Song |
| National Institutes of Health | R21EB030072 | Yike Wang<br>Matthew R Lowerison<br>Qi You<br>Bing-Ze Lin<br>Daniel A Llano<br>Pengfei Song |

| Funder | Grant reference number | Author |
|---|---|---|
| National Institutes of Health | R21EB030072-01S1 | Yike Wang<br>Matthew R Lowerison<br>Qi You<br>Bing-Ze Lin<br>Daniel A Llano<br>Pengfei Song |
| National Institutes of Health | R21AG077173 | Yike Wang<br>Matthew R Lowerison<br>Qi You<br>Bing-Ze Lin<br>Daniel A Llano<br>Pengfei Song |

The funders had no role in study design, data collection, and interpretation, or the decision to submit the work for publication.

## Author contributions

Yike Wang, Conceptualization, Data curation, Software, Validation, Investigation, Visualization, Methodology, Writing – original draft, Writing – review and editing; Matthew R Lowerison, Conceptualization, Methodology, Writing – review and editing; Zhe Huang, Qi You, Bing-Ze Lin, Data curation, Methodology, Writing – review and editing; Daniel A Llano, Conceptualization, Supervision, Investigation, Methodology, Writing – original draft, Writing – review and editing; Pengfei Song, Conceptualization, Supervision, Funding acquisition, Investigation, Methodology, Writing – original draft, Project administration, Writing – review and editing

## Author ORCIDs

Yike Wang ⓘ https://orcid.org/0009-0003-0113-6014
Matthew R Lowerison ⓘ http://orcid.org/0000-0002-1125-4554
Zhe Huang ⓘ http://orcid.org/0000-0002-3477-8326
Bing-Ze Lin ⓘ http://orcid.org/0000-0001-7323-4236
Daniel A Llano ⓘ https://orcid.org/0000-0003-0933-1837
Pengfei Song ⓘ https://orcid.org/0000-0002-9103-6345

## Ethics

All experimental procedures were conducted in accordance with the guidelines set by the University of Illinois Institutional Animal Care and Use Committee (IACUC Protocol number #22165).

Reviewer #1 (Public review): https://doi.org/10.7554/eLife.95168.4.sa1
Reviewer #2 (Public review): https://doi.org/10.7554/eLife.95168.4.sa2
Author response https://doi.org/10.7554/eLife.95168.4.sa3

# Additional files

## Supplementary files

MDAR checklist

## Data availability

The code for the analyses presented in this paper is openly accessible at https://github.com/eker-wang/ULMQuantitativeAnalysis (copy archived at *Wang, 2025*). Raw IQ data associated in this study are available at Zenodo (https://doi.org/10.5281/zenodo.17393921).

The following dataset was generated:

| Author(s) | Year | Dataset title | Dataset URL | Database and Identifier |
|---|---|---|---|---|
| Wang Y, Lowerison M, Huang Z, You Q, Lin B-Z, Llano D, Song P | 2025 | Longitudinal Awake Imaging of Mouse Deep Brain Microvasculature with Super-resolution Ultrasound Localization Microscopy | https://doi.org/10.5281/zenodo.17393921 | Zenodo, 10.5281/zenodo.17393921 |

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
