## [Editor Report · eLife Assessment]

This study presents **important** methodologies for repeated brain ultrasound localization microscopy (ULM) in awake mice and a set of results indicating that wakefulness reduces vascularity and blood flow velocity. The data supporting these findings are **solid**. This study is relevant for scientists investigating vascular physiology in the brain.

---

## [Referee Report · Reviewer #1 (Public review)]

Summary:

Wang and Colleagues present a study aimed at demonstrating the feasibility of repeated ultrasound localization microscopy (ULM) recording sessions on mice chronically implanted with a cranial window transparent to US. They provided quantitative information on their protocol, such as the required number of Contrast enhancing microbubbles (MBs) to get a clear image of the vasculature of a brain coronal section. Also, they quantified the co-registration quality over time-distant sessions and the vasodilator effect of isoflurane.

Strengths:

Strengths:

The study showed a remarkable performance in recording precisely the same brain coronal section over repeated imaging sessions. In addition, it sheds light on the vasodilator effect of isoflurane (an anesthetic whose effects are not fully understood) on the different brain vasculature compartments, although, as the Authors stated, some insights in this aspect have already been published with other imaging techniques. The experimental setting and protocol are very well described.

In this newly revised version, the Authors made evident efforts to strengthen the messages of their study. All the limitations of their research have been clearly acknowledged.

A central issue remains. To answer my concerns about the need for multivariate analyses, the Author stated that: "Due to the limited number of animals used, the analyses presented in this work should be interpreted as example case studies." Although this sentence does not convince me, if the purpose of this study was to showcase the potentialities of ULM for future longitudinal awake studies, why don't they avoid any statistics? The trend for decreased vein size and increased arterial blood flow during wakefulness is evident from the plot and physiologically plausible. Why impose wrong statistics instead of dropping them altogether? I do not see the lack of statistics as detrimental to this study, based on the feedback received from the Authors.

---

## [Referee Report · Reviewer #2 (Public review)]

Summary:

The authors present a very interesting collection of methods and results using brain ultrasound localization microscopy (ULM) in awake mice. They emphasize the effect of the level of anesthesia on the quantifiable elements assessable with this technique (i.e. vessel diameter, flow speed, in veins and arteries, area perfused, in capillaries) and demonstrate the possibility of achieving longitudinal cerebrovascular assessment in one animal during several weeks with their protocol.

The authors made a good rewriting the article based on the reviewers' comments. One of the message of the first version of the manuscript was that variability in measurements (vessel diameter, flow velocity, vascularity) were much more pronounced under changes of anesthesia than when considering longitudinal imaging across several weeks. This message is now not quite mitigated, as longitudinal imaging seems to show a certain variability close to the order of magnitude observed under anesthesia. In that sense, the review process was useful in avoiding hasty conclusion and calls for further caution in ULM awake longitudinal imaging, in particular regarding precision of positioning and cancellation of tissue motion.

Strengths:

Even if the methods elements considered separately are not new (brain ULM in rodents, setup for longitudinal awake imaging similar to those used in fUS imaging, quantification of vessel diameters/bubble flow/vessel area), when masterfully combined as it is done in this paper, they answer two questions that have been long-running in the community: what is the impact of anesthesia on the parameters measured by ULM (and indirectly in fUS and other techniques)? Is it possible to achieve ULM in awake rodents for longitudinal imaging? The manuscript is well constructed, well written, and graphics are appealing.

The manuscript has been much strengthened by the round of review, with more animals for the longitudinal imaging study.

Weaknesses:

The manuscript has been only marginally modified since our last round of review, so there is probably not much we reviewers can additionally elaborate to improve it. Therefore my last concerns about the reliability of longitudinal quantifications and on certain discrepancies remains for this paper. As a general piece of advice, I would just say that every claim (' is higher', is lower', is stable') should be supported by evidence and statistical testing if it is not already the case.

Response 06: the authors' response is not satisfactory. Even if the difference in terms of ROI boundaries between fig 4e and fig 4j has been underlined by the authors, they only provide a wordy comment and no additional quantitative analysis that could explain the discrepancy I pointed out. By doing so they take the risk of making misinterpretations. The reader is left with a discrepancy that could be explained by 2 mechanisms: -pial vessel population behave differently from penetrating arterioles and venules OR - the imaging of pial vessels with ULM is not good enough to enable proper quantification because the vessels are not clearly visible (out of plane extent). In any case Figure 4j does not "provides a more comprehensive representation of cortical vasculature" as stated. If the changes in pial vessels cannot be reliably measured, they should be excluded from the ROI.

Line 161: be careful with the use of vessel density, as pointed by reviewer 1.

Line 196: "the decrease in venous vessel area (averaging 55% across mice) was greater than that of arterial (averaging 35%)" no stat test has been performed.

---

## [Author Response]

The following is the authors’ response to the previous reviews

**Public Reviews:**

**Reviewer #1 (Public Review):**
Summary:Wang and Colleagues present a study aimed at demonstrating the feasibility of repeated ultrasound localization microscopy (ULM) recording sessions on mice chronically implanted with a cranial window transparent to US. They provided quantitative information on their protocol, such as the required number of Contrast enhancing microbubbles (MBs) to get a clear image of the vasculature of a brain coronal section. Also, they quantified the co-registration quality over time-distant sessions and the vasodilator effect of isoflurane.Strengths:The study showed a remarkable performance in recording precisely the same brain coronal section over repeated imaging sessions. In addition, it sheds light on the vasodilator effect of isoflurane (an anesthetic whose effects are not fully understood) on the different brain vasculature compartments, although, as the Authors stated, some insights in this aspect have already been published with other imaging techniques. The experimental setting and protocol are very well described.Wang and co-authors submitted a revised version of their study, which shows improvements in the clarity of the data description.However, the flaws and limitations of this study are substantially unchanged.The main issues are:Statistics are still inadequate. The TOST test proposed in this revised version is not equivalent to an ANOVA. Indeed, multivariate analyses should be the most appropriate, given that some quantifications were probably made on multiple vessels from different mice. The 3 reviewers mentioned the flaws in statistics as the primary concern.

Response 01: We thank the reviewer for raising this important point. We fully acknowledge the limitations of our current statistical analysis. We would like to clarify that the TOST procedure was applied exclusively to the measurements taken from the same vessel segment in the same animal across different time points, with the purpose of evaluating the consistency of vessel diameter measurements. We recognize that the statistical analysis in this study remains limited, which we have acknowledged as a key limitation in the manuscript. This constraint arises primarily from the limited number of animals, and our analysis should be interpreted as a representative case study rather than a generalized statistical conclusion. We have revised the manuscript to clarify these points and to more explicitly acknowledge the statistical limitations.

(Line 329) “Our current study primarily focused on demonstrating the feasibility of longitudinal ULM imaging in awake animals, instead of conducting a systematic investigation of how isoflurane anesthesia alters cerebral blood flow. Due to the limited number of animals used, the analyses presented in this work should be interpreted as example case studies. While the trends observed across animals were consistent, the small sample size restricts the scope of statistical inference. For future work, it would be valuable to design more rigorous control experiments with larger sample sizes to systematically compare the effects of isoflurane anesthesia, awake states, and other anesthetics that do not induce vasodilation on cerebral blood flow.”

No new data has been added, such as testing other anesthetics.

Response 02: We acknowledge that the current study does not include data involving other anesthetics, and we have also discussed this point in our initial response. In fact, we did attempt to use other anesthetics such as ketamine. However, we found it difficult to draw reliable conclusions due to experimental limitations such as variable anesthesia recovery profiles and injection timing, as elaborated in the following paragraphs. Therefore, we decided not to include these data in the current study to avoid potential misinterpretation.

One major limitation of our experimental setup is that imaging in the awake state is necessarily conducted after a brief period of isoflurane-anesthesia. This brief anesthesia allows for the intravenous injection of microbubbles via the tail vein. Isoflurane is particularly suited for this purpose due to its rapid onset and offset. Mice can recover quickly once the gas is withdrawn, which enables relatively consistent post-anesthesia imaging in the awake state.

In contrast, other anesthetic agents present challenges. Their recovery profiles are slower, more variable, and less controllable. Reversal drugs can be administered to awaken the animals, but they add another variability. These may lead to greater fluctuations in cerebral hemodynamics and factors introduce uncertainty in the timing of bolus microbubble injection. As such, our current setup is not ideal for systematically comparing different anesthetics and could yield misleading results.

A more appropriate strategy for comparing awake ULM imaging with different anesthetics would be performing awake imaging first, followed by imaging under anesthesia. This would ensure that the awake condition is free from residual anesthetic effects. However, this method raises higher requirement in bubble delivery, as no anesthesia can be used for the intravenous injection.

To address this, we are actively exploring another solution using indwelling jugular vein catheterization. By surgically implanting a catheter into the jugular vein prior to imaging, we can establish a stable and reproducible route for microbubble delivery in fully awake animals without any anesthesia induction. This method has the potential to enable direct and reliable comparisons across different physiological states. However, the implementation of this technique and the associated experimental findings go beyond the scope of the current study and will be presented in a future manuscript.

In the present work, we have emphasized the methodological limitations of our approach and clarified that our primary goal is to highlight the necessity and feasibility of awake-state ULM imaging. The focus is not to comprehensively characterize the effects of different anesthetic agents on microvascular brain flow. We appreciate your understanding and interest in this important future direction.

Based the responses and previous revision, we have further refined the discussion of the relevant limitations:

(Line 324) “Although isoflurane is widely used in ultrasound imaging because it provides long-lasting and stable anesthetic effects, it is important to note that the vasodilation observed with isoflurane is not representative of all anesthetics. Some anesthesia protocols, such as ketamine combined with medetomidine, do not produce significant vasodilation and are therefore preferred in experiments where vascular stability is essential, such as functional ultrasound imaging. Our current study primarily focused on demonstrating the feasibility of longitudinal ULM imaging in awake animals, instead of conducting a systematic investigation of how isoflurane anesthesia alters cerebral blood flow. Due to the limited number of animals used, the analyses presented in this work should be interpreted as example case studies. While the trends observed across animals were consistent, the small sample size restricts the scope of statistical inference. For future work, it would be valuable to design more rigorous control experiments with larger sample sizes to systematically compare the effects of isoflurane anesthesia, awake states, and other anesthetics that do not induce vasodilation on cerebral blood flow.”

(Line 347) “Another limitation of this study is the potential residual vasodilatory effect of isoflurane anesthesia on awake imaging sessions and the short imaging window available after bolus injection. The awake imaging sessions were conducted shortly after the mice had emerged from isoflurane anesthesia, required for the MB bolus injections. The lasting vasodilatory effects of isoflurane may have influenced vascular responses, potentially contributing to an underestimation of differences in vascular dynamics between anesthetized and awake state. In addition, since microbubbles are rapidly cleared from circulation, the duration of effective imaging is limited to only a few minutes, which also overlaps with the anesthesia recovery period, constraining the usable awake-state imaging window. Future improvement on microbubble infusion using an indwelling jugular vein catheter presents a promising alternative to address these limitations. This method allows for stable microbubble infusion without the need for anesthesia induction, ensuring that the awake imaging condition is free from residual anesthetic effects. Moreover, it has the potential to extend the duration of imaging sessions, offering a longer and more stable time window for data acquisition. Furthermore, by performing ULM imaging in the awake state first, instead of starting with anesthetized imaging, researchers can achieve a more rigorous comparison of how various anesthetics influence cerebral microvascular dynamics relative to the awake baseline.”

The Authors still insist on using the term Vascularity which they define as: 'proportion of the pixel count occupied by blood vessels within each ROI, obtained by binarizing the ULM vessel density maps and calculating the percentage of the pixels with MB signal.'. Why not use apparent cerebral blood volume or just CBV? Introducing an unnecessary and redundant term is not scientifically acceptable. In this revised version, vascularity is also used to indicate a higher vascular density (Line 275), which does not make sense: blood vessels do not generate from the isoflurane to the awake condition in a few minutes. Rev2 also raised this point.

Response 03: Thank you for revisiting this important point. We acknowledge that the term vascularity is difficult to interpret for readers, and we also recognize that we did not sufficiently justify its use in the earlier version.

Based on your suggestion, we have now replaced all instances of “vascularity” with “fractional vessel area”. While the underlying definition remains the same, fractional vessel area offers a more intuitive description. The term “fractional” denotes that the vessel area is normalized to the total area of the selected ROI. This normalization is essential for fair comparisons across ROIs of different sizes, such as Figures 4i–k to evaluate various brain regions. We would also like to clarify that this was not introduced as an unnecessary or redundant term, but rather as a more suitable metric for longitudinal ULM analysis. We did consider using apparent cerebral blood volume (CBV), estimated from microbubble counts. However, we found that it was less robust and meaningful in the context of longitudinal ULM comparisons. Below we provide further justification for using the vessel area instead:

(1) Using the vessel area is more robust:

In longitudinal ULM comparisons, normalization across time points is essential to enable fair and meaningful comparisons. In our study, we normalized the data based on a cumulative 5 million microbubbles (e.g., Fig. 2). Other normalization strategies could also be adopted, as long as the resulting vascular maps reach a sufficiently saturated state. However, even with normalization, it remains important to use a quantitative metric that is minimally biased and invariant to experimental fluctuations across time points. Vessel area, derived from binarized vessel maps, is less sensitive to variations in acquisition time and microbubble concentration. This is because repeated microbubble trajectories through the same location are not counted multiple times. In contrast, apparent CBV, calculated from the microbubble counts, is more susceptible to different concentration conditions. Since repeated detections in the same location accumulate, the metric can be dependent on injection efficiency and imaging duration. While CBV may still be valid under well-controlled, steady-state conditions, we found the vessel area to be a more robust and reliable metric for longitudinal analysis under our current bolus-injection protocol.

(2) Using the vessel area is more meaningful:

Compared to CBV, the vessel area provides a more direct representation of structural characteristics such as vessel diameter. Anesthesia-induced vasodilation leads to an increase in vessel diameter. Although local diameter changes can be assessed by manually selecting vessel segments, this approach is labor-intensive and prone to selection bias. To enable a more comprehensive and objective assessment of such morphological changes, fractional vessel area provides a more informative alternative to CBV, as it captures diameter-related variations at a global or regional scale, and avoids potential biases associated with manually selecting specific vessels or regions.

In response to: vascularity is also used to indicate a higher vascular density (Line 275), which does not make sense: blood vessels do not generate from the isoflurane to the awake condition in a few minutes.

We agree that blood vessels cannot be generated in a few minutes. Vascularity (now fractional vessel area) should be interpreted as apparent vessel density, which reflects a probabilistic estimate of vessel density based on the detectable microbubble.

Both apparent vessel density and apparent CBV are indirect, sampling-based approximations of vascular features, and both are fundamentally limited by microbubble detection sensitivity. Low microbubble concentrations lead to underestimation of both CBV and vessel area. A change from zero to non-zero in these metrics does not imply the physical appearance or disappearance of vessels, but rather reflects a change in the likelihood of detecting flow in each region.

In summary, while neither fractional vessel area (vascularity in previous versions) nor apparent CBV is a perfect metric due to the inherent limitations of ULM, we believe the vessel area provides a more robust and meaningful parameter for our longitudinal comparisons. We have revised the main text to include this explanation and acknowledge the limitations and interpretation of fractional vessel area more explicitly.

Revision in Results:

(Line 181) “To validate the broader applicability of our findings, we conducted ROI-based analyses using fractional vessel area and mean velocity as primary metrics. These metrics extended the analysis of vessel diameter and flow velocity to entire brain regions or selected ROIs, which provides a more objective assessment of cerebral blood flow changes at a global scale and reduces the bias associated with manually selecting vessel segments. For vessel area measurements, the term fractional denotes that the vessel area is normalized to the total area of the selected ROI. This normalization is essential for fair comparisons across ROIs of different sizes.”

Revision in Methods: definition of vascularity

(Line 571) “In ROI-based analysis, we focused on two primary parameters: fractional vessel area and mean velocity. Fractional vessel area was defined as the proportion of the pixel count occupied by blood vessels within each ROI, obtained by binarizing the ULM vessel density maps and calculating the percentage of the pixels with MB signal. Mean velocity was calculated by averaging all non-zero pixel of velocity estimates within the ROI. The velocity distribution within each ROI was also visualized using violin plots, as shown in Fig. 2, 4 and 6, to illustrate the range and density of flow velocity estimates across different acquisition. In this study, we focused on these two metrics because they represent the most straightforward extension of single-vessel analysis to brain-wide vascular changes.”

We put our ROI analysis code on GitHub and added a “Code availability” section. We hope it can serve as a foundation for users to explore different quantitative metrics in their own longitudinal ULM studies. We hope to provide an example to inspire further exploration.

(Line 578) “Code availability

To support quantitative longitudinal analysis of ULM data, we developed an open-source MATLAB application (https://github.com/ekerwang/ULMQuantitativeAnalysis). This tool is designed to facilitate ROI-based analysis of ULM images for longitudinal comparisons. It supports multiple quantification metrics, including but not limited to vessel area and mean velocity used in this study. Users can select and adapt different metrics based on their specific applications, as a wide range of ULM-based quantification metrics have been developed for different pathological and pharmacological studies.”

The long-term recordings mentioned by the Authors refer to the 3-week time frame analyzed in this study. However, within each acquisition, the time available from imaging is only a few minutes (< 10', referring to most of the plots showing time courses) after the animals' arousal from isoflurane and before bubbles disappear. This limitation should be acknowledged.

Response 04: Thank you for this comment. We agree that the current imaging sessions are constrained by the short time window available after the animal’s arousal from isoflurane and before bubbles disappear. This limitation indeed restricts the duration of usable awake-state imaging in our current bolus injection protocol. As discussed earlier, we are actively exploring the use of a jugular vein catheterization approach to address this limitation. This approach has the potential to extend the imaging session duration and provide a longer, more stable time window. We have now acknowledged this limitation more explicitly in the revised Discussion section.

(Line 347) “Another limitation of this study is the potential residual vasodilatory effect of isoflurane anesthesia on awake imaging sessions and the short imaging window available after bolus injection. The awake imaging sessions were conducted shortly after the mice had emerged from isoflurane anesthesia, required for the MB bolus injections. The lasting vasodilatory effects of isoflurane may have influenced vascular responses, potentially contributing to an underestimation of differences in vascular dynamics between anesthetized and awake state. In addition, since microbubbles are rapidly cleared from circulation, the duration of effective imaging is limited to only a few minutes, which also overlaps with the anesthesia recovery period, constraining the usable awake-state imaging window. Future improvement on microbubble infusion using an indwelling jugular vein catheter presents a promising alternative to address these limitations. This method allows for stable microbubble infusion without the need for anesthesia induction, ensuring that the awake imaging condition is free from residual anesthetic effects. Moreover, it has the potential to extend the duration of imaging sessions, offering a longer and more stable time window for data acquisition. Furthermore, by performing ULM imaging in the awake state first, instead of starting with anesthetized imaging, researchers can achieve a more rigorous comparison of how various anesthetics influence cerebral microvascular dynamics relative to the awake baseline.”

The more precise description of the number of mice and blood vessels analyzed in Figure 6 makes it apparent the limited number of independent samples used to support the findings of this work. A limitation that should be acknowledged. The newly provided information added as Supplementary Figure 1 should be moved to the main text, eventually in the figure legends. The limited data in support of the findings was also highlighted by Rev2 and, indirectly, by Rev3.

Response 05: We acknowledge the limited number of independent samples used in this study. In the revised manuscript, we have explicitly emphasized this limitation in the Discussion section. Specifically, we added the following statement:

(Line 329) “Our current study primarily focused on demonstrating the feasibility of longitudinal ULM imaging in awake animals, instead of conducting a systematic investigation of how isoflurane anesthesia alters cerebral blood flow. Due to the limited number of animals used, the analyses presented in this work should be interpreted as example case studies. While the trends observed across animals were consistent, the small sample size restricts the scope of statistical inference. For future work, it would be valuable to design more rigorous control experiments with larger sample sizes to systematically compare the effects of isoflurane anesthesia, awake states, and other anesthetics that do not induce vasodilation on cerebral blood flow.”

Following your suggestion, we have also moved the newly provided information (the table in Supplementary Figure 1) into figure captions. In addition, we have modified in the Methods section to ensure that this information is clear.

(Line 406) “Eight healthy female C57 mice (8-12 weeks) were used for this study, numbered as Mouse 1 to Mouse 8. Three mice (Mouse 1–3) were used to compare imaging results between awake and anesthetized states (Fig. 3 and 4). Three additional mice (Mouse 4–6) underwent longitudinal imaging over a three-week period (Fig. 5 and 6). Among them, Mouse 4 was also used as an example to demonstrate the overall system schematic and saturation conditions (Fig. 1 and 2). Several mice (Mouse 2, 6, 7, and 8) exhibited suboptimal cranial window quality or image artifacts and were included to illustrate common surgical or imaging issues (Supplementary Fig. 1). The specific usage of each animal is also annotated in the corresponding figure captions.”

**Reviewer #2 (Public Review):**
The authors present a very interesting collection of methods and results using brain ultrasound localization microscopy (ULM) in awake mice. They emphasize the effect of the level of anesthesia on the quantifiable elements assessable with this technique (i.e. vessel diameter, flow speed, in veins and arteries, area perfused, in capillaries) and demonstrate the possibility of achieving longitudinal cerebrovascular assessment in one animal during several weeks with their protocol.The authors made a good rewriting of the article based on the reviewers' comments. One of the message of the first version of the manuscript was that variability in measurements (vessel diameter, flow velocity, vascularity) were much more pronounced under changes of anesthesia than when considering longitudinal imaging across several weeks. This message is now not quite mitigated, as longitudinal imaging seems to show a certain variability close to the order of magnitude observed under anesthesia. In that sense, the review process was useful in avoiding hasty conclusion and calls for further caution in ULM awake longitudinal imaging, in particular regarding precision of positioning and cancellation of tissue motion.Strengths:Even if the methods elements considered separately are not new (brain ULM in rodents, setup for longitudinal awake imaging similar to those used in fUS imaging, quantification of vessel diameters/bubble flow/vessel area), when masterfully combined as it is done in this paper, they answer two questions that have been longrunning in the community: what is the impact of anesthesia on the parameters measured by ULM (and indirectly in fUS and other techniques)? Is it possible to achieve ULM in awake rodents for longitudinal imaging? The manuscript is well constructed, well written, and graphics are appealing.The manuscript has been much strengthened by the round of review, with more animals for the longitudinal imaging study.Weaknesses:Some weaknesses remain, not hindering the quality of the work, that the authors might want to answer or explain.When considering fig 4e and fig 4j together: it seems that in fig 4e the vascularity reduction in the cortical ROI is around 30% for downward flow, and around 55% for upward flow; but when grouping both cortical flows in fig 4j, the reduction is much smaller (~5%), even at the individual level (only mouse 1 is used in fig 4e). Can you comment on that?

Response 06: Thank you for carefully pointing this out. This discrepancy arises primarily from differences in ROI selections.

The vascularity metric (now we changed the term into fractional vessel area, based on Reviewer 1’s comments) is calculated as the proportion of vessel-occupied pixels relative to the total ROI area. As such, it is best suited for longitudinal comparisons within the same ROI rather than across-ROI comparisons, particularly when the size and vessel composition of the ROIs differ.

In Fig. 4e, the cortical ROI includes mostly the penetrating vessels, which are selected due to their clear distinction between upward (venous) and downward (arterial) flow directions. Pial vessels were intentionally excluded because flow direction alone does not reliably distinguish arteries from veins in these surface vessels. Thus, the goal of this analysis was to indicate arteriovenous differences, rather than to represent the full cortical vascular changes.

In contrast, the ROIs used in Fig. 4j aim to provide a more comprehensive view of cortical vascular responses without distinguishing flow direction. That’s why both penetrating and pial vessels are included. Since pial vessels showed relatively smaller vascularity changes within the coronal cross-sections analyzed in our study, their inclusion in the cortical ROI likely contributed to the smaller overall reduction in vascularity observed in Figure 4j.

To address this potential confusion, we have added further clarification in the Results section of the revised manuscript.

(Line 209) “It is worth noting that prior analyses (Fig. 4d–h) aimed to illustrate arteriovenous differences. Since pial vessels are difficult to distinguish as arteries or veins based on flow direction in coronal plane imaging, they were excluded from the ROI selection in those analyses. In the current whole-brain comparisons (Fig. 4i-k), the cortical ROIs no longer exclude pial vessels, since distinguishing between arteries and veins is not required. This aims to provide a more comprehensive representation of cortical vasculature.”

When considering fig 4e, fig 4j, fig 6e and fig 6i altogether, it seems that vascularity can be highly variable, whether it be under anesthesia or vascular imaging, with changes between 5 to 40%. Is this vascularity quantification worth it (namely, reliable for example to quantify changes in a pathological model requiring longitudinal imaging)?

Response 07: Thank you for raising this important point. We found that imaging in the awake state is inherently more variable than under anesthesia. In contrast, anesthetized imaging offers a more controlled and stable physiological condition, as anesthesia suppresses many sources of variation. For pathological studies, if the vascular or hemodynamic changes induced by anesthesia do not interfere with the scientific question being addressed, imaging under anesthesia can still be a practical and effective approach, due to its experimental simplicity and better physiological consistency.

The higher variability observed in awake imaging arises from both physiological fluctuations in animals and unavoidable experimental inconsistencies, such as small misalignment on the imaging plane across sessions. If the research question aims to avoid the confounding effects of anesthesia, then instead of suppressing variation through anesthesia, it is important to acknowledge the natural baseline variation in the awake state. However, efforts should be made to minimize technical sources of variation. We have added a brief discussion of this issue at the end of the manuscript to reflect this consideration.

(Line 396) “However, it is also important to note that although longitudinal awake imaging presents promise to avoid the confounding effects of anesthetics, imaging under anesthesia remains more convenient and controllable in many cases. For applications where the physiological question of interest is not sensitive to anesthesia-induced vascular effects, anesthetized imaging still offers a simpler and more stable approach. Awake imaging inherently exhibits greater physiological variability. However, care must be taken at the experimental level to minimize confounding sources of variation, such as stress level of the animal or handling inconsistencies, to ensure that the measurements are physiologically meaningful.”

Regarding whether fractional vessel area (formerly referred to as vascularity) is a worthwhile metric for longitudinal quantification: based on our experience and comparisons, we found vessel area to be relatively robust and informative (see also Response 02 to Reviewer 1 for details). However, we acknowledge that other quantitative metrics—such as microbubble count, tortuosity, or flow directionality—may be more suitable depending on the specific pathological model or research question. How these metrics perform in awake imaging and longitudinal disease models is indeed an open and important question. We hope our work can serve as a foundation to inspire further investigation in this direction. To facilitate such exploration, we have developed and open-sourced a MATLAB-based analysis tool that supports multiple quantitative ULM metrics for longitudinal comparison. We encourage users to adapt and extend this framework to evaluate different quantitative metrics.

(Line 578) “Code availability

To support quantitative longitudinal analysis of ULM data, we developed an open-source MATLAB application (https://github.com/ekerwang/ULMQuantitativeAnalysis). This tool is designed to facilitate ROI-based analysis of ULM images for longitudinal comparisons. It supports multiple quantification metrics, including but not limited to vessel area and mean velocity used in this study. Users can select and adapt different metrics based on their specific applications, as a wide range of ULM-based quantification metrics have been developed for different pathological and pharmacological studies.”

**Reviewer #2 (Recommendations For The Authors):**
Images in figure 4 lack color bars.

Response 08: Thank you for pointing this out. The color bars for the images in Figure 4 are the same as those used in the corresponding images in Figure 3. We have now added the explanation of color bars to the revised version of Figure 4 caption.

Fig 4d: upward and downward are probably swapped.

Response 09: Thank you for pointing this out, and we apologize for the oversight. They were mistakenly swapped. We have corrected this error in the revised figure.

No quantitative conclusions are drawn regarding the changes in vessel diameter under anesthesia? Is it not significant? If it is not then why bring changes in diameter to our attention in fig 3 (white arrows) and figure 4b?

Response 10: Our intention in highlighting diameter changes in Figure 3 (white arrows) and Figure 4b was to provide an illustrative example of isoflurane-induced diameter changes at the single-vessel level. These examples are meant to serve as case studies, not as the basis for broad statistical conclusions.

In the initial version of the manuscript, we attempted to draw quantitative conclusions by measuring vessel diameters from ten manually selected vessel segments at each location. However, based on feedback from other reviewers, we decided to remove this analysis in the revised version. Manual selection of vessel segments is highly subjective and prone to bias, limiting its reliability for quantitative interpretation.

Instead, we focused on ROI-based analysis using fractional vessel area (formerly referred to as vascularity), which reflects widespread changes in vessel diameter across regions. It is a more generalizable and less biased metric for quantifying vascular diameter changes.

We further explained this in the Results section:

(Line 181) “To validate the broader applicability of our findings, we conducted ROI-based analyses using fractional vessel area and mean velocity as primary metrics. These metrics extended the analysis of vessel diameter and flow velocity to entire brain regions or selected ROIs, which provides a more objective assessment of cerebral blood flow changes at a global scale and reduces the bias associated with manually selecting vessel segments. For vessel area measurements, the term fractional denotes that the vessel area is normalized to the total area of the selected ROI. This normalization is essential for fair comparisons across ROIs of different sizes.”

Line 210 "In summary, statistical analysis revealed a decrease in individual vessel diameter" this does not seem to be supported by this version of the manuscript as no analysis is done on a representative group of vessels for the diameter.

Response 11: Thank you for pointing out this important issue. In line with our previous response (Response 10), we would like to clarify that the analysis of individual vessel diameter was intended to serve as an example study, rather than a statistically supported conclusion based on a group of vessels. To avoid confusion, we have removed the phrase “statistical analysis revealed a decrease in individual vessel diameter” from the manuscript.

The meaning of the *** in fig 6b and 6c should be clarified as: -it is not explicitly stated - the equivalence test interpretation is less usual than other tests.

Response 12: We thank the reviewer for pointing out this important issue. We agree that the use of asterisks (***) in Fig. 6b and 6c may have led to confusion, as such markers are typically associated with statistical significance in difference testing. In our case, the analysis was based on the two one-sided test (TOST) procedure to assess statistical equivalence, which is indeed less commonly used and could be misinterpreted.

To address this, we have replaced the asterisks *** in the figure with the label “equiv.”, which more clearly reflects the intended interpretation. Additionally, we have revised the figure caption and the main text to explicitly state that these markers denote statistical equivalence (not difference) as determined by TOST, with the equivalence margin defined as three times the standard deviation of one week.

(Figure 6 Caption) “Statistical analysis was performed using the two one-sided test (TOST) to evaluate consistency of measurement. The label “equiv.” indicates statistically equivalent measurements (p < 0.001), defined as interweek differences smaller than three times the standard deviation of one week.”

(Line 240) “Statistical testing of equivalence was conducted using the two one-sided test (TOST) procedure, which evaluates whether the difference between two time points falls within a predefined equivalence margin. Specifically, equivalence is defined as the inter-week difference being smaller than three times the standard deviation of one week. A statistically significant result in TOST (p < 0.001) supports the interpretation that the measurements are statistically equivalent, which is denoted as “equiv.” in the figures.”

Line 237 and following: please consider rephrasing into "To further generalize these findings and examine longitudinal variation in ROI-based analysis, we used Mouse 4 as an example to show the consistency of blood flow density across different flow directions in the cortex (Fig. 6d) and extended the quantitative analysis to all three mice (Fig. 6e) (individual ULM upward and downward flow images for all three mice over the threeweek longitudinal study period can be found in Supplementary Fig. 4)." The paragraph will make much more sense.

Response 13: We appreciate your helpful rephrasing. We have fully adopted your proposed revision to enhance the clarity and coherence of the text. The sentence now reads exactly as you recommended:

(Line 250): “To further generalize these findings and examine longitudinal variation in ROI-based analysis, we used Mouse 4 as an example to show the consistency of blood flow density across different flow directions in the cortex (Fig. 6d) and extended the quantitative analysis to all three mice (Fig. 6e) (individual ULM upward and downward flow images for all three mice over the three-week longitudinal study period can be found in Supplementary Fig. 4).”

Line 248: "While arterial and venous flow velocity distributions exhibit clear distinctions, their variations over the three weeks remained acceptable" the meaning of acceptable remains elusive.

Response 14: Thank you for pointing out the ambiguity in the phrase “remained acceptable”. To improve clarity and precision, we have revised the sentence to provide a more informative description. The updated sentence now reads:

(Line 261) “While arterial and venous flow velocity distributions exhibit clear distinctions, the distribution shapes remained relatively consistent across the three weeks. Specifically, variation in median velocity were within 1 mm/s. In contrast, anesthesia-induced changes can lead to velocity shifts exceeding 1 mm/s.”

Line 253: consider rephrasing in "Despite subcortical regions showing the largest vascularity variability consecutive to anesthesia-induced changes, vascularity in those regions was relatively stable values in the longitudinal study" as otherwise the link between the 2 parts of the sentence feels odd.

Response 15: Thank you for your constructive suggestion regarding the logical flow of the sentence. We fully agree with your point and have revised the sentence exactly as you proposed.

(Line 268) “Despite subcortical regions showing the largest vascularity variability consecutive to anesthesia-induced changes, vascularity in those regions was relatively stable values in the longitudinal study.”